



# Photolytically Induced Changes in Composition and Volatility of Biogenic Secondary Organic Aerosol from Nitrate Radical Oxidation during Night-to-day Transition

Cheng Wu[1], David M. Bell[2*], Emelie L. Graham[1], Sophie Haslett[1], Ilona Riipinen[1], Urs Baltensperger[2], Amelie Bertrand[2], Stamatios Giannoukos[2,a], Janne Schoonbaert[2], Imad El Haddad[2], Andre S. H. Prevot[2], Wei Huang[3], Claudia Mohr[1*]

[1]Department of Environmental Science, Stockholm University, Sweden

[2]Laboratory of Atmospheric Chemistry, Paul Scherrer Institute, Villigen, Switzerland

[3]Institute for Atmospheric and Earth System Research / Physics, Faculty of Science, University of Helsinki, Helsinki, 00014, Finland

[a]Now at: Department of Chemistry and Applied Biosciences, ETH Zurich, Switzerland

*Correspondence to*: David M. Bell (*david.bell@psi.ch*) and Claudia Mohr (*claudia.mohr@aces.su.se*)

**Abstract.** Night-time reactions of biogenic volatile organic compounds (BVOCs) and nitrate radicals ($NO_3$) can lead to the formation of secondary organic aerosol ($BSOA_{NO3}$). Here we study the impacts of light exposure on the chemical composition and volatility of $BSOA_{NO3}$ formed in the dark from three precursors (isoprene, α-pinene, and β-caryophyllene) in atmospheric simulation chamber experiments. Our study represents $BSOA_{NO3}$ formation conditions where reactions between peroxy radicals ($RO_2 + RO_2$) and between $RO_2$ and $NO_3$ are favored. The emphasis here is on the identification of particle-phase organonitrates (ONs) formed in the dark and their changes during photolytic aging on timescales of ~1 h. Chemical composition of particle-phase compounds was measured with a chemical ionization mass spectrometer with filter inlet for gases and aerosols (FIGAERO-CIMS) and an extractive electrospray ionisation time-of-flight mass spectrometer (EESI-TOF). Volatility information of $BSOA_{NO3}$ was derived from FIGAERO-CIMS desorption profiles (thermograms) and a volatility tandem differential mobility analyser (VTDMA). During photolytic aging, there was a relatively small change in mass due to evaporation (< 5 % for the isoprene and α-pinene $BSOA_{NO3}$, 12 % for the β-caryophyllene $BSOA_{NO3}$), but we observed significant changes in the chemical composition of the $BSOA_{NO3}$. Overall, 53 %, 45 %, and 62 % of the total mass for the isoprene, α-pinene, and β-caryophyllene $BSOA_{NO3}$ was sensitive to photolytic aging and exhibited decay. The photolabile compounds include both monomers and oligomers. Oligomers can decompose into their monomer units through photolysis of the bonds (e.g. likely O-O) between them. Fragmentation of both oligomers and monomers also happened at other positions, causing the formation of compounds with shorter carbon skeletons. The cleavage of the nitrate functional group from the carbon chain was likely not a main degradation pathway in our experiments. In addition, photolytic degradation of compounds changes their volatility, and can lead to evaporation. We used different methods to assess bulk volatilities and discuss their changes during both dark aging and photolysis in context of the chemical changes we observed. We also reveal large





uncertainties in saturation vapor pressure estimated from parameterization for the ON oligomers with multiple nitrate groups. Overall, our results suggest that photolysis causes photodegradation of a substantial fraction of BSOA$_{NO3}$, changes both the chemical composition and the bulk volatility of the particles, and might be a potentially important loss pathway of BSOA$_{NO3}$

during the night-to-day transition.

## 1 Introduction

Secondary organic aerosol (SOA), formed via the oxidation of volatile organic compounds (VOCs) emitted from human activities (anthropogenic) and vegetation (biogenic), has important impacts on climate (Shrivastava et al., 2017) and human

health (Daellenbach et al., 2020). Biogenic VOCs such as isoprene ($C_5H_8$), monoterpenes ($C_{10}H_{16}$) and sesquiterpenes ($C_{15}H_{24}$), are key precursors for global SOA formation due to their larger emissions (Guenther et al., 2006; Guenther et al., 1995) and higher reactivity towards atmospheric oxidants compared to the VOCs from anthropogenic emissions. While oxidation initiated by ozone ($O_3$) and hydroxyl radicals (OH) dominates during daytime, nitrate radicals ($NO_3$), generated at night by the reaction of nitrogen dioxide ($NO_2$) with $O_3$, are the major nocturnal oxidant. Modeling studies estimate that 5–21 % of SOA

are produced globally by $NO_3$ chemistry (Hoyle et al., 2007; Pye et al., 2010).

The reaction of $NO_3$ with VOCs is a major pathway for the production of organonitrates (ONs, $RONO_2$), which represent a substantial fraction of submicron aerosol nitrate at both urban and rural sites (Kiendler-Scharr et al., 2016). ONs also play an important role in the removal and transport of nitrogen oxides ($NO_x$), and impact $NO_x$ cycling and $O_3$ formation (Perring et al., 2013). While the lifetime of aerosols in the atmosphere typically spans over multiple day/night cycles, the lifetime of ONs

within the particles is much shorter (Lee et al., 2016; Zare et al., 2018). Reactions influencing particulate lifetime of ONs include oxidation, hydrolysis (Pye et al., 2015; Takeuchi and Ng, 2019), and photolysis (Müller et al., 2014; Suarez-Bertoa et al., 2012). They change both chemical and physical properties of SOA and ONs, and have different impacts on the atmospheric $NO_x$ budget.

Compared to OH and $O_3$, there are few laboratory studies on $NO_3$-initiated biogenic SOA (BSOA$_{NO3}$). Very little is known

about the chemical composition and volatility of BSOA$_{NO3}$. While there are extensive studies on photolytic/photochemical (photolysis + OH) aging of OH- and $O_3$- initiated SOA under low $NO_x$ conditions (Surratt et al., 2006; Walser et al., 2007; Mang et al., 2008; Pan et al., 2009; Henry and Donahue, 2012; Presto et al., 2005; Zawadowicz et al., 2020), studies on photolytic/photochemical aging of $NO_3$-initiated SOA and ONs are limited. The few studies that have examined optical properties of SOA show that there is a significant difference in SOA produced from $NO_3$ oxidation compared to other oxidation

pathways (Nakayama et al., 2015; Peng et al., 2018), as well as in BSOA$_{NO3}$ from different precursors (He et al., 2021). Nah et al. (2016) reported that SOA from β-pinene + $NO_3$ is generally resilient to photochemical aging and does not exhibit significant changes in its chemical composition, while most of the SOA from α-pinene + $NO_3$ evaporates during photochemical aging (photolysis + OH). However, little is known about SOA from other VOC precursors, and formed in chemical regimes





with different branching ratios for the peroxy radical ($RO_2$) pathways such as $RO_2 + HO_2$, $RO_2 + NO_3$ and $RO_2 + RO_2$. The knowledge gaps related to the effects of photolysis on lifetime and physicochemical properties of ONs currently hinders a quantitative understanding of their impacts on the atmospheric nitrogen budget, chemical interactions between anthropogenic and biogenic compounds, and climate.

In this study, we performed chamber experiments of dark formation and photolytic aging of $NO_3$-initiated SOA and ONs from three biogenic VOCs, namely isoprene ($C_5H_8$), α-pinene ($C_{10}H_{16}$), and β-caryophyllene ($C_{15}H_{24}$). This article is a companion paper to "Particle-phase processing of α-pinene $NO_3$ secondary organic aerosol in the dark", published in the same journal (Bell et al., 2021). Bell et al. (2021) present the evolution of the composition of $NO_3$-initiated SOA from α-pinene during dark aging. Here, we focus on the impacts of photolytic aging on both the chemical composition and volatility of $BSOA_{NO3}$. We compare the chemical composition of particle-phase compounds before and after 1 h of ultraviolet (UV) irradiation using advanced mass spectrometric techniques, determine photolabile fractions for each $BSOA_{NO3}$ system, and investigate the potential chromophores that photolyze. In addition, we also examine the changes in the bulk volatility with both parameterization and experimental methods.

## 2 EXPERIMENTAL SECTION

### 2.1 Chamber and experimental description

Experiments were conducted in the Paul Scherrer Institute (PSI) Simulation Chamber for Atmospheric Chemistry (PSI-SCAC, (Platt et al., 2013)), which is an 8 $m^3$ flexible Teflon chamber suspended in a temperature-controlled shipping container. The chamber is surrounded by a bank of black lights (40 x 100 W Cleo Performance solarium lamps, Philips (Krapf et al., 2016)). The experiments were performed under humid conditions (RH ≈ 60 %) and at a temperature of 21 ± 3 °C (with an increase of 3–4 °C during photolysis). A list of the experiments is given in Table 1.

**Table 1 List of experiments and summary of experimental conditions**

| Exp. | Precursor | VOC reacted (ppb) | $N_2O_5$ (ppb) | Max OA loading ($\mu g\ m^{-3}$) | SOA yield (%) |
|---|---|---|---|---|---|
| 1 | Isoprene | 100 | 180–220 | 24 | 9 |
| 2 | Isoprene | 100 | 100–120 | 11 | 4 |
| 3 | α-Pinene | 100 | 300 | 18 | 3 |
| 4 | β-Caryophyllene | 50 | 200+ | 464 | 110 |

SOA was formed in the chamber by the reaction of the precursors with $NO_3$ radicals. Each experiment followed a similar protocol. First, the chamber was cooled from 30 °C to ~20 °C, then humidity was increased by boiling MilliQ water (18 M-Ohm) until the desired RH was reached. After the chamber conditions were stabilized, the desired VOC was injected into the chamber volumetrically, and its concentration was monitored with a proton transfer reaction mass spectrometer (Quadrupole-PTR-MS, Ionicon). β-Caryophyllene could not be monitored with the PTR-MS because its mass-to-charge ($m/z$) ratio of 204



Th is outside of the mass transmission range for quantitative measurements, though trace amounts of the molecule were detected in the experiment. After the VOC concentration had been stable for 5–15 min, $N_2O_5$ was injected into the chamber, which acted as the source of $NO_3$ radicals via decomposition. $N_2O_5$ was synthesized by reacting $O_3$ (~1–2 %, with an Innovatec ozone generator using PSI-provided $O_2$) with a pure source of $NO_2$ (99 %). $N_2O_5$ crystals were collected by passing the gaseous

components into a cooled flask (-70 °C). Each day the sample was brought to the chamber in a dry-ice ethanol bath and warmed for 1–2 min prior to injection (5 L min$^{-1}$ for ~10 s). A 0-d box model (F0AM) (Wolfe et al., 2016) utilizing the Master Chemical Mechanism (MCM) (Jenkin et al., 1997) was used to model the VOC precursor decay to determine a range of initial $N_2O_5$ concentrations. This was performed when the VOC was measurable, which was not the case for the β-caryophyllene experiments (Exp. 4), where the estimate of 200+ ppb (Table 1) comes from the average $N_2O_5$ concentrations in the other

experiments performed. After the injection was complete, the chamber was stirred by injecting zero air into the chamber (100 L min$^{-1}$ for ~10 min). Following chamber stirring, the volume of the chamber started to decrease due to all of the instruments sampling air from the chamber. Experiments started under dark conditions to probe $NO_3$-initiated SOA formation, and to follow its chemical transformation with no external stimulus. After ~3–5 h of dark aging, the chamber was irradiated with UV lights ($\lambda_{Max}$ = 350 nm, see Platt et al. (2013)) for about 1 h to test the impacts of photolytic aging. After each experiment, ~1000

ppb $O_3$ was added, the chamber was continuously flushed with zero air (~100 L min$^{-1}$), and the temperature of the enclosure was increased to 30 °C overnight to promote the evaporation of $HNO_3$ from the walls of the chamber.

O$_3$ was measured with an $O_3$ analyzer (Thermo 49C), $NO_x$ and $NO_2$ with a chemiluminescence $NO_x$ monitor (Thermo 42C). Aerosol number size distributions were measured by a scanning mobility particle sizer (SMPS, TSI model 3938). Both a chemical ionization mass spectrometer with filter inlet for gases and aerosols (FIGAERO-CIMS, Aerodyne Research, Inc.)

with iodide as the reagent ion, and an extractive electrospray ionization time-of-flight mass spectrometer (EESI-TOF, Tofwerk) were used to measure the molecular composition of organic compounds. The FIGAERO-CIMS performed semi-continuous online measurements and shifted analysis between the particle phase and the gas phase, and the EESI-TOF continuously measured the particle phase with rapid response. In a subset of the experiments (Exp. 1, Exp. 2 and Exp. 4), the evaporation of particles was measured with a custom-built volatility tandem differential mobility analyser (VTDMA) (Tritscher et al.,

115  2011).

## 2.2 Chemical ionization mass spectrometer with filter inlet for gases and aerosols

The design and operation of the FIGAERO-CIMS were similar to those described in previous studies (Lopez-Hilfiker et al., 2014; Lee et al., 2014; Huang et al., 2018). In this study, the FIGAERO inlet was coupled to a high-resolution time-of-flight chemical-ionization mass spectrometer (HR-ToF-CIMS) ($M/\Delta M$ ~5,000–6,000), and I$^-$ was used as reagent ion. An X-ray

generator was used to ionize methyl iodide and produce the reagent ion in a nitrogen flow. Particles were collected on a 25mm Zefluor® PTFE filter (Pall Corp.) inside the FIGAERO via a sampling port (stainless steel tube of ca. 1.5 m length, flow rate 5 L min$^{-1}$). The duration of particle-phase sampling depended on the mass concentrations in the chamber and was 10–20 min for most of the experiments. For each experiment, three to four filters were sampled during dark aging, and one filter was



collected after about 1 h of photolytic aging. In addition, one particle blank was performed for each experiment during dark
aging (details about background subtraction see Fig. S1 and the supporting information (SI)). During particle-phase collection,
gases were measured via a 6 mm Teflon tube of ~1 m length at 5 L min$^{-1}$. When the particle-phase sampling was completed,
the gas-phase measurement was switched off and particles on the filter were desorbed by a flow of ultra-high-purity (99.999
%) nitrogen. A FIGAERO desorption round lasted about 55 min: 20 min of ramping the temperature of the nitrogen flow from
ambient temperature up to 200 °C were followed by a 20 min "soak period" at a constant temperature of 200 °C, and 15 min
of cooling down to room temperature. The mass spectral signal evolutions as a function of desorption temperature are termed
thermograms (Lopez-Hilfiker et al., 2014). The integration of thermograms of individual compounds (cooling period excluded)
yields their total signal in counts per deposition. Here we did not convert the counts per deposition into chamber concentrations,
as we only show normalized signal (either signal normalized to the dominating compound or to the sum signal of all
compounds). We observed modest or negligible fragmentation due to thermal desorption in the FIGAERO inlet (5–27 %, 1–4
%, 10–23 % of the total organic signal of the isoprene, α-pinene and β-caryophyllene SOA, respectively). Artefacts resulting
from thermal fragmentation products were corrected (see the SI and Fig. S2). Thermal fragmentation products were detected
either through thermograms of individual compounds with multiple peaks (normally double peaks) or one peak with $T_{max}$
(desorption temperature at which a compound′s signal exhibits a maximum) much higher than the estimated $T_{max}$ (e.g., for α-
pinene SOA, a $C_{10}$ monomer had a $T_{max} \approx 140$ °C which is in the range of $T_{max}$ for $C_{20}$ dimers (Faxon et al., 2018)). In addition,
only a small fraction (< 1 % of the total mass) of the compounds were deprotonated. In this study, we only report the molecules
clustered with I$^-$. Further information concerning the correction of thermal decomposition and the data analysis is provided in
the SI.

**2.3 Extractive electrospray ionization time-of-flight mass spectrometer**

A detailed description of the EESI-TOF can be found in previous work (Lopez-Hilfiker et al., 2019; Pospisilova et al., 2020)
and the companion paper (Bell et al., 2021). In brief, the EESI-TOF samples the aerosol formed in the chamber via a sampling
line (~3.5 m long stainless steel) and removes the gaseous components by passing the sampled volume through a multi-channel
charcoal denuder. The aerosol sample then intersects a spray of droplets emanating from a fused silica electrospray capillary
(50:50 H$_2$O:acetonitrile doped with 100 ppm NaI). The droplets envelope the aerosol sample and extract the water soluble
fraction of the aerosol. During droplet evaporation a majority of the extracted molecules bind to Na$^+$, creating positively
charged adducts. The ions are guided to the time-of-flight mass spectrometer ($M/\Delta M$ ~5,500–7,000) where their mass-to-
charge ratio is determined. Background measurements were conducted every 4 minutes for 1 minute by passing the sample air
through a particle filter. The reported signal throughout the work are the difference spectra of consecutive 4-minute average
signal and 1-minute average background signal. The EESI-TOF signal (Hz) is scaled by the molecular weight of each molecular
ion in order to represent the EESI-TOF signal as a mass-based measurement (ag s$^{-1}$). Artefacts associated with these
measurements are discussed in the companion paper associated with this publication (Bell et al. 2021). The EESI-TOF data





for most experiments had similar bulk sensitivities, except for the α-pinene experiment (Exp. 3), which was about a factor of 10 lower because the EESI-ToF was still being optimized, and capillary position and TOF settings were being altered.

**2.4 Volatility tandem differential mobility analyser**

The custom-built VTDMA (see e.g. Tritscher et al. (2011)) sampled aerosols through the same inlet as the EESI-TOF and the
SMPS, and a silica gel diffusion drier downstream from the chamber. The VTDMA combines two differential mobility particle sizer (DMPS) systems, coupled in series with a heating unit in between. Both DMPS systems consist of a custom made DMA (Stockholm University, operated with closed-loop sheath air) and a condensation particle counter (CPC, TSI 3010). Before entering each DMPS system the particles were brought into charge equilibrium using a Ni-63 source. The first DMA selected a nearly monodisperse aerosol with the diameter set to the geometric mean diameter measured in situ by the SMPS upstream
(122, 84, and 208 nm during both dark aging and photolysis for Exp. 1, Exp. 2 and Exp. 4, respectively). The sample flow was then split into two, with one half going to the CPC measuring the selected particle concentration and the other half entering a 35 cm long custom-built thermodenuder (TD) with a residence time of 1.9 s, which heated the aerosol to 150 °C in Exp. 1 and Exp. 2 and 175 °C in Exp. 4. The size distribution (over 15 bins) of the heated aerosol was measured in the second DMPS system with a time resolution of approximately 5 min, and the change in size was converted to an estimate of the volume
fraction remaining (VFR):

$$\text{VFR} = \frac{D(T)^3}{D_{\text{init.}}^3}$$
(Eq. 1)

where $D_{\text{init.}}$ is the initial diameter, and $D(T)$ are the heated (Temperature $T$) diameters, represented by the geometric mean mode diameter estimated using a Gaussian fit of the whole heated size distribution.

**2.5 Wall loss correction**

The particle mass concentrations (in μg m$^{-3}$) were derived from integrated number size distributions and their conversions to mass using an assumed organic aerosol density (1.19 g cm$^{-3}$ (Vaden et al., 2011)). The particle mass concentration and the ion signals of the particle-phase compounds measured by the FIGAERO-CIMS and the EESI-TOF were corrected for coagulation and wall losses using a uniform dynamic wall loss rate $k_{\text{wall}}$ for the whole size range:

$$\frac{dN}{dt} = -k_{\text{coag}} * N^2 - k_{\text{wall}} * N$$
(Eq. 2)

$k_{\text{wall}}$ is determined from the observed exponential decay of the particle number concentration, which is corrected for coagulation following Eq. 2, where $N$ corresponds to the particle number concentration and $k_{\text{coag}}$ corresponds to the coagulation coefficient ($5*10^{-10}$ s$^{-1}$ (Pospisilova et al., 2020)). In the experiments performed here, the total number concentration (in cm$^{-3}$) varied between $10^6$ and $10^4$ at the beginning of the experiment to $10^4$ and $10^3$ at the end of the experiment. In experiments with large number concentrations (e.g. > $5*10^4$ cm$^{-3}$, in Exp. 4), coagulation was a dominant particle sink, which resulted in a mobility
diameter that increased during the experiment, even though the mass was decreasing from evaporation. The particle number concentration was constrained to be constant throughout the lifetime in the chamber, based upon $k_{\text{wall}}$ and $k_{\text{coag}}$. Evaporation of





semi-volatile compounds will not affect the particle wall loss correction because this does not change the particle number concentration, only the particle volume (or mass).

**2.6 Estimating bulk saturation mass concentration $C^*$ based on FIGAERO-CIMS and VTDMA**

The saturation mass concentration ($C^*$) of individual compounds was calculated based on their molecular composition using three different previously published parameterizations:

1) an updated version of the parameterization by Donahue et al., 2011, modified based on the saturation concentrations of highly oxygenated molecules (HOMs) detected by Tröstl et al. (2016) and published by Mohr et al. (2019):

$$\log_{10}C^* = (n_0 - n_C)b_C - (n_O - 3n_N)b_O - 2\frac{n_C(n_O - 3n_N)}{n_C + n_O - 3n_N}b_{CO} - n_N b_N \quad \text{(Eq. 3)}$$

where $n_0 = 25$, $b_C = 0.475$, $b_O = 0.2$, $b_{CO} = 0.9$ and $b_N = 2.5$. $n_C$, $n_O$ and $n_N$ are number of carbon, oxygen and nitrogen atoms in the compound, respectively. For ONs, it is assumed that a nitrate group reduces a compound´s vapor pressure by about 2.5 orders of magnitude (Donahue et al., 2011; Pankow and Asher, 2008).

2) an updated version of Li et al. (2016) with modified nitrogen coefficient for organic nitrates (Isaacman-VanWertz and Aumont, 2020):

$$\log_{10}C^* = (n_0 - n_C)b_C - n_O b_O - 2\frac{n_C n_O}{n_C + n_O}b_{CO} - n_N b_N \quad \text{(Eq. 4)}$$

where $n_0 = 22.66$, $b_C = 0.4481$, $b_O = 1.656$, $b_{CO} = -0.7790$ for CHO compounds, and $n_0 = 24.13$, $b_C = 0.3667$, $b_O = 0.7732$, $b_{CO} = -0.07790$ and $b_N = -1.5464$ for CHON compounds. $n_C$, $n_O$ and $n_N$ are number of carbon, oxygen and nitrogen atoms in the compound, respectively. The modification of the parameterization is based on the observation that a nitrate group has a similar impact on vapor pressure as a hydroxyl, thus one nitrate group (one nitrogen atom and three oxygen atoms) reduces a
compound's vapor pressure by 0.7732 orders of magnitude (Isaacman-VanWertz and Aumont, 2020).

3) a parameterization based on highly oxygenated organic molecules (HOMs) from α-pinene ozonolysis (Peräkylä et al. (2020)):

$$\log_{10}C^* = 0.18 \times n_C - 0.14 \times n_H - 0.38 \times n_O + 0.80 \times n_N + 3.1 \quad \text{(Eq. 5)}$$

where $n_C$, $n_H$, $n_O$ and $n_N$ are number of carbon, hydrogen, oxygen and nitrogen atoms in the compound, respectively. With this
parameterization, a nitrate group reduces a compound's vapour pressure by 0.34 orders of magnitude.

We note here that more such parametrizations exist in literature (Donahue et al., 2011; Daumit et al., 2013). We choose these three parameterizations as they have explicit formulations for nitrate groups/nitrogen. The parametrizations calculate $C^*$ at a temperature of 300 K. The temperature dependence of $C^*$ is considered as follows:

$$C^*(T) = C^*(300\,\text{K}) \exp\left(\frac{\Delta H^{VAP}}{R}\left(\frac{1}{300\,\text{K}} - \frac{1}{T}\right)\right) \quad \text{(Eq. 6)}$$

where $T$ is the temperature in Kelvin, $C^*(300\,\text{K})$ is the saturation concentration at 300 K, $R$ is the gas constant, and $\Delta H^{VAP}$ is the vaporization enthalpy:

$$\Delta H^{VAP} = -11\log_{10}C^*(300\,\text{K}) + 129\,; \Delta H^{VAP} < 200\,\text{kJ/mol} \quad \text{(Epstein et al., 2010)} \quad \text{(Eq. 7)}$$



For each filter sample from the FIGAERO-CIMS, the mass-weighted average $\log_{10}C^*$ for the entire particle population deposited on the filter (bulk) was calculated and adjusted to the same $T$ (298.3 K) as that used by the kinetic model (see the description below) for further comparison.


The $T_{max}$, the desorption temperature at which a compound's signal exhibits a maximum in the FIGAERO-CIMS, has earlier been shown to be connected qualitatively to the compound's volatility ($\log_{10}C^*$ or enthalpy of vaporization) (Lopez-Hilfiker et al., 2014; Thornton et al., 2020; Bannan et al., 2019). For each filter sample from the FIGAERO-CIMS, we calculated the mass-weighted average $T_{max}$ of the entire particle population deposited on the filter (bulk).

As an additional method to constrain the bulk volatility, VTDMA measurements of the VFR were used to determine the bulk $\log_{10}C^*$ with a kinetic model that simulates the evaporation of a monodisperse aerosol in the heated part of the VTDMA (Riipinen et al., 2010). The bulk $\log_{10}C^*$ (at 298.3 K) was determined manually so that the calculated VFR matches the measurements. Additional model input parameters include the settings of the VTDMA, i.e. initial particle diameter and concentration, together with the residence time and temperature in the heated section and the ambient temperature at the time

of the measurement.

## 3 RESULTS AND DISCUSSION

### 3.1 Mass yields of isoprene, α-pinene and β-caryophyllene BSOA$_{NO3}$

As shown in Table 1, the wall loss-corrected SOA yields and maximal mass concentrations formed from isoprene, α-pinene, and β-caryophyllene are 4–9 % at 11–24 µg m$^{-3}$, 3 % at 18 µg m$^{-3}$ and 110 % at 464 µg m$^{-3}$, respectively. The SOA yields

observed here are comparable to previously reported values: For isoprene + NO$_3$, yields between 2 % and 24 % have been presented (Rollins et al., 2009; Ng et al., 2008); for α-pinene + NO$_3$, between 0 % and 16 % (Hallquist et al., 2009; Nah et al., 2016; Fry et al., 2014); and for β-caryophyllene + NO$_3$, between 86 % and 146 % (Fry et al., 2014; Jaoui et al., 2013). The yields at the lower end of the range of reported values of the isoprene and α-pinene SOA in our study can likely be explained by the lack of seed aerosol in those experiments. For β-caryophyllene, most of the products are of sufficiently low volatility

(see section 3.5) and the SOA yields are similar to those from the seeded experiments. The variation in SOA yields can also be caused by different chemical regimes, resulting e.g. in different branching ratios for the peroxy radicals (RO$_2$) + hydroperoxyl radical (HO$_2$), RO$_2$ + NO$_3$ and RO$_2$ + RO$_2$ pathways after the initial RO$_2$ formation via NO$_3$ + VOCs. The RO$_2$ + RO$_2$ pathway is a more effective channel for forming SOA than the RO$_2$ + HO$_2$ and RO$_2$ + NO$_3$ pathways (Ng et al., 2008). In Ng et al. (2008), the SOA yield from the isoprene + NO$_3$ experiments (seeded), where the RO$_2$ + RO$_2$ pathway dominated,

was much higher (23.8 %) than that from experiments where the RO$_2$ + NO$_3$ pathway dominated (4.3 %), because RO$_2$ + RO$_2$ chemistry led to the formation of oligomers. In the study of Nah et al. (2016), the branching ratios for the RO$_2$ + HO$_2$, RO$_2$ + NO$_3$ and RO$_2$ + RO$_2$ pathways were 5:4:1, and a SOA yield of 1.7 % at 1.2 µg m$^{-3}$ (no seed) was observed for α-pinene + NO$_3$ SOA. In our experiments, we achieved higher SOA yields from α-pinene + NO$_3$, due to N$_2$O$_5$:VOC ratios between 1 and 5 and the radical balance being dominated by either RO$_2$ + RO$_2$ (at low N$_2$O$_5$:VOC) or RO$_2$ + NO$_3$ (at high N$_2$O$_5$:VOC) chemistry

and no significant source of HO$_2$ radicals. As discussed in Bell et al. (2021), even in experiments where the ratio of N$_2$O$_5$:VOC is high (~3) and the RO$_2$ + NO$_3$ pathway is predicted by the MCM model to be dominant, the initial composition of BSOA$_{NO3}$ formed here in the dark is still dominated by RO$_2$ + RO$_2$ reactions, resulting in substantial fractions of oligomers and indicating that the rate of the RO$_2$ + RO$_2$ pathway is likely underestimated by the MCM model.

The evolution of the SOA mass concentrations and particle size during both dark aging (2 h before photolysis) and photolysis

are shown in Fig. 1 for all systems. Before photolysis, for the α-pinene system (Exp. 3), steady evaporation was observed from the particles, which is consistent with shrinking mean diameter (from 240 to 233 nm) resulting in 15 % loss of particle mass (wall-loss corrected) or volume. For the two isoprene experiments (Exp. 1 and Exp. 2), ~6 % of the mass was lost during dark aging. The wall-loss corrected mass concentrations of the β-caryophyllene SOA are stable throughout the dark aging period, while the geometric mean diameter of the particles increases from 229 nm (at time -2 h in Fig. 1) to 266 nm (at time 0 h), as a

result of particle coagulation following the large number concentrations (~10$^6$ cm$^{-3}$). The differences in evaporation can likely be attributed to the lower volatility of β-caryophyllene monomers (C$_{15}$) compared to isoprene dimers (C$_{10}$) and α-pinene monomers (C$_{10}$) (for details see section 3.5). Overall, the BSOA$_{NO3}$ formed during dark aging exhibits slight to moderate shrinkage of its mass and size (after wall loss-correction) after the initial production.

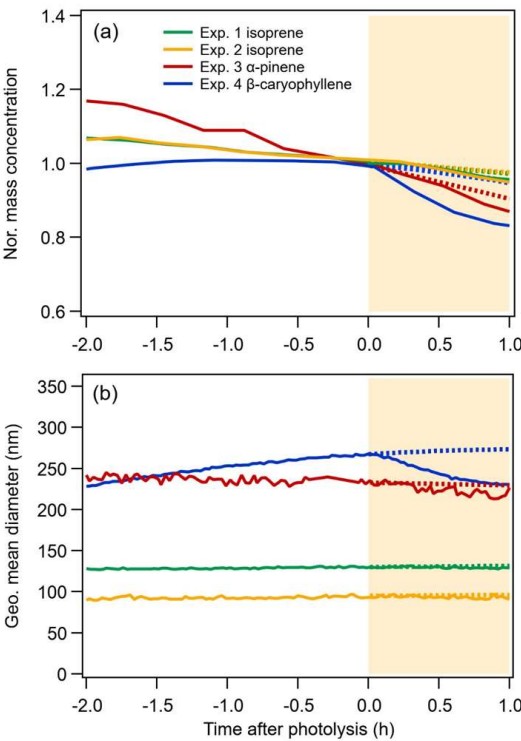





**Fig. 1 Time series of normalized mass concentration (a) and geometric mean diameter (b) of the SOA measured by the SMPS. SOA mass loadings were corrected for particle wall losses based on the decay of the particle number concentration. The mass concentration is normalized to that when the lights were turned on. The area shaded in yellow corresponds to the period of UV irradiation. The expected curvature of the mass concentration and geometric mean diameter without UV irradiation is represented by the dashed lines.**

**3.2 Chemical composition of isoprene, α-pinene and β-caryophyllene BSOA$_{NO3}$**

In total, 478, 425, and 366 organic molecular compositions (including 436, 332, and 348 ONs) were identified with the FIGAERO-CIMS, and 359, 158, and 441 organic molecular compositions (including 273, 134, and 268 ONs) were identified with the EESI-TOF, for the isoprene, α-pinene and β-caryophyllene SOA, respectively. In the α-pinene + NO$_3$ experiment (Exp. 3), because of the lower sensitivity of the EESI-TOF compared to other experiments (described in 2.3), the detected

compounds by the EESI-TOF were fewer than that detected in other experiments. Due to the dominating RO$_2$ + RO$_2$ pathway, oligomers make a substantial contribution to the total mass formed in the dark. The mass fractions of oligomers are ~100 %, 86–88 % and 60–63 % with the FIGAERO-CIMS and 97–98 %, 85–99 %, and 13–22 % with the EESI-TOF, for the isoprene, α-pinene and β-caryophyllene SOA, respectively. The dimer-to-monomer ratio varies across the three systems, and it increases with decreasing carbon number of the precursors.

The mass spectra of the particle-phase species of the three systems obtained from the FIGAERO-CIMS and the EESI-TOF shortly before switching on the lights are shown in Fig. 2. For the isoprene system, Exp. 1 and Exp.2 are similar, thus only Exp. 1 is presented here and Exp. 2 is presented in the SI (Fig. S5). For the isoprene SOA, monomers are negligible (C$_{1-5}$ compounds are less likely to partition into the particle phase due to their relatively high volatility), and dimers with three to four nitrate groups (e.g. C$_{10}$H$_{17}$N$_3$O$_{13}$ and C$_{10}$H$_{18}$N$_4$O$_{16}$, each contributing 15 % to the total mass) dominate the mass measured

by the FIGAERO-CIMS. Dimers with one, two and five nitrate groups, and trimers (C$_{15}$) also contribute a substantial fraction (65 %) to the total mass. The major compounds observed in our study are similar to those observed by Ng et al. (2008), who used a UPLC/(–)ESI-TOFMS and investigated ratios of N$_2$O$_5$:isoprene from 0.6 to 5. For the EESI-TOF, the major compounds are similar, with the same highest peak (C$_{10}$H$_{17}$N$_3$O$_{13}$) as measured by the FIGAERO-CIMS. However, the relative intensities are different. Compared to the FIGAERO-CIMS, the EESI-TOF measures higher signals of C$_{10}$H$_{16}$N$_2$O$_{10}$, C$_{15}$H$_{24}$N$_4$O$_{18}$ and

C$_{20}$H$_{25}$N$_3$O$_{19}$, but lower signals of C$_{10}$H$_{17}$N$_3$O$_{12}$, C$_{10}$H$_{18}$N$_4$O$_{16}$ and C$_{10}$H$_{17}$N$_5$O$_{18}$. There is a high signal from C$_4$H$_6$O$_2$, which comes from the degradation of oligomeric species C$_{10}$H$_{16}$N$_2$O$_{9,10}$ and C$_{10}$H$_{17}$N$_3$O$_{13}$, to be discussed in a future paper (Bell et al., in preparation).



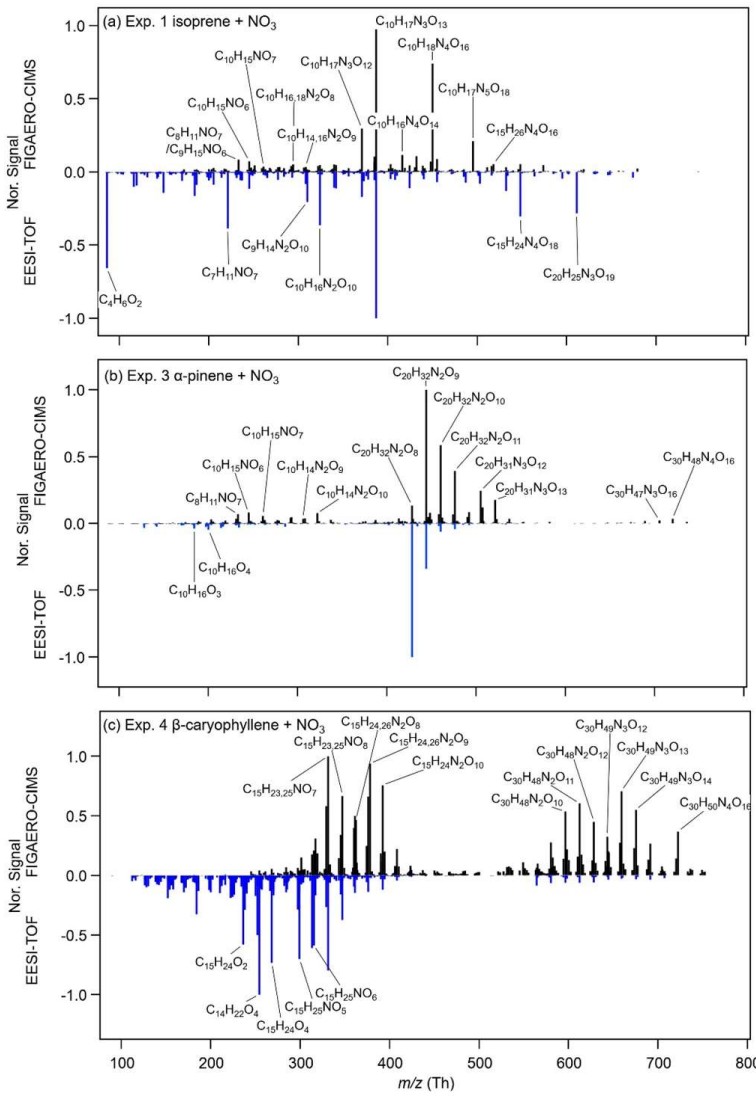

**Fig. 2** FIGAERO-CIMS (positive axis) and EESI-TOF (negative axis) mass spectra of particle phase $C_xH_yO_zN_{0-4}$ formed during (a)
**Exp.1: isoprene + $NO_3$, (b) Exp. 3: α-pinene + $NO_3$ and (c) Exp. 4: β-caryophyllene + $NO_3$ shortly before photolysis. For FIGAERO-CIMS, the last filter before photolysis is chosen (Pre 2 in the Fig. 3). The EESI-TOF mass spectra are averaged mass spectra during the same sampling time as that of FIGAERO-CIMS. All mass spectra are normalized to the corresponding maximal signal.**





The α-pinene system according to the FIGAERO-CIMS measurements is dominated by $C_{20}$ dimers with two nitrate groups, i.e. $C_{20}H_{32}N_2O_{8-13}$ make up 39 % of the total mass, while the monomers and trimers contribute 12–14 % and 3.7–4.4 %,

respectively. Our mass spectra are similar to those reported in Takeuchi and Ng (2019) measured with a FIGAERO-CIMS, where the α-pinene + $NO_3$ SOA ($N_2O_5$:α-pinene ~7) was dominated by $C_{20}$ dimers with two dominating compounds, $C_{20}H_{32}N_2O_9$ and $C_{20}H_{32}N_2O_{10}$. The EESI-TOF detected similar dimers and monomers as those measured by the FIGAERO-CIMS. However, the highest peak measured is $C_{20}H_{32}N_2O_9$ for the FIGAERO-CIMS and $C_{20}H_{32}N_2O_8$ for the EESI-TOF. It is also worth noting that the EESI-TOF measured a few CHO compounds without nitrate groups, such as $C_{10}H_{16}O_3$ and $C_{10}H_{16}O_4$,

which may also stem from some degradation pathways in the EESI-TOF source.

The β-caryophyllene SOA mass spectrum from the FIGAERO-CIMS is dominated by $C_{15}$ monomers with 1–2 nitrate groups ($C_{15}H_{23,25}NO_{7,8}$ and $C_{15}H_{24,26}N_2O_{8-10}$) and $C_{30}$ dimers with 2–3 nitrate groups ($C_{30}H_{48}N_2O_{10-12}$ and $C_{30}H_{49}N_3O_{12-14}$). The mass spectrum from the EESI-TOF is dominated by $C_{15}$ monomers with zero and one nitrate group. The contribution from the dimers is substantially smaller compared to FIGAERO-CIMS, which may result from a decreased sensitivity of larger $m/z$ components

as observed previously (Lopez-Hilfiker et al., 2019) and/or from decomposition of ON dimers. The higher contribution from the monomers, compared to the isoprene and α-pinene systems, is likely due to the larger number of carbon atoms of β-caryophyllene, and the two double bonds (one more than α-pinene), which both lead to low volatility of the monomers (for details see section 3.5).

Overall, both instruments are able to cover the major compounds from both monomer and dimer regions, although the

sensitivity towards different compounds varies. The I⁻ CIMS is not sensitive towards non-oxygenated compounds, monoalcohols, monoketones or monoaldehydes (Lee et al., 2014), while the EESI-TOF is sensitive to nearly all compounds present in SOA but not sensitive to non-oxygenated compounds (Lopez-Hilfiker et al., 2019). Some difference may also be caused by the fragmentation of oligomers into smaller compounds ($C_xH_yO_2$) with the EESI-TOF, likely occurring due to the proximity of $–ONO_2$ functional groups next to peroxy linkages which results in fragmentation when exposed to water in the

electrospray (Bell et al, in preparation), and a loss of a $HNO_3$ fragment during ionization (Liu et al., 2019).

### 3.3 Photolytic aging of isoprene, α-pinene and β-caryophyllene BSOA$_{NO3}$

As can be observed in Fig. 1, switching on the lights has effects on particle mass and size for all BSOA$_{NO3}$ systems, but with different extent. For the isoprene SOA, the particle size and mass show a slightly steeper decrease compared to the situation expected without UV irradiation (dashed lines). For the α-pinene SOA, both the particle size and mass decrease compared to

the dark, from 229 nm and a normalized mass concentration of 0.9 to 219 nm and 0.87 after 1 h photolysis. The β-caryophyllene SOA, although the least volatile, shows a decrease by 12 % (normalized mass concentration of 0.83 versus 0.95 expected without UV irradiation), and a shrinking of the mean particle size from 273 nm to 230 nm. These results indicate that there are slight to moderate losses of particle phase compounds due to UV irradiation. The possible reasons causing the differences in the three systems will be discussed in this section and in section 3.5.




In Fig. 3, the relative changes in the chemical composition of the SOA before (pre) and after (post) photolytic aging measured by the FIGAERO-CIMS are illustrated. During irradiation, processes/reactions occurring in the dark may also continue with the lights on, e.g. consumption/decomposition of reactive oxygen species (Pospisilova et al., 2020), evaporation of semi-volatile compounds, and hydrolysis of nitrate functional groups can alter the chemical composition of SOA particles. To account for this, Fig. 3 compares the chemical composition of two particle filter samples collected during dark aging (Pre 1

and Pre 2), and one filter sample after 1 h of photolytic aging (Post). The time interval of the sampling was about 1.5–2 h or about 3 h if a particle blank was performed between two filters. The difference between the two filters during dark aging (Pre 2 – Pre 1) shows the changes of the chemical composition caused by all other processes than photolysis, while the difference between the last filter during dark aging and the one after 1 h photolysis (Post – Pre 2) shows the impacts of the above-mentioned processes plus the impact caused by UV irradiation.

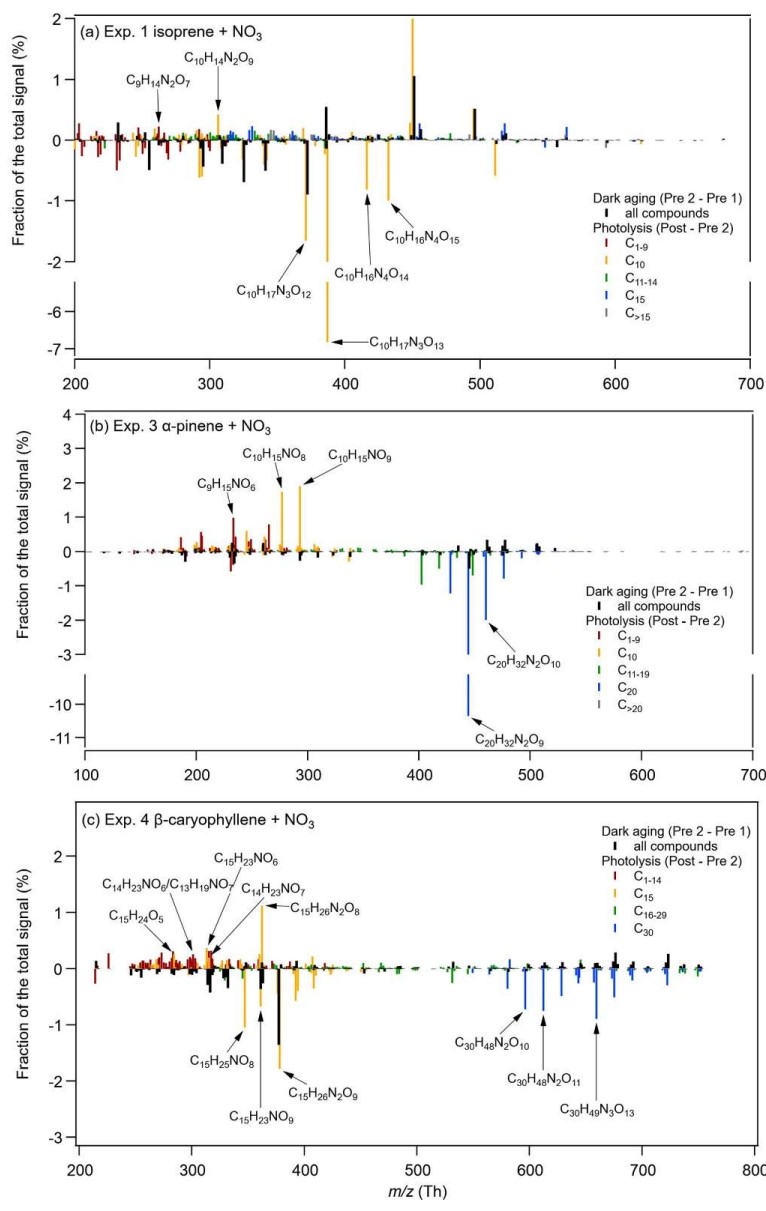


**Fig. 3 Relative changes of the fraction of the total signal between two filters during dark aging (Pre 2 – Pre 1) and between the last filter during dark aging and one filter after 1 h of photolysis (Post – Pre 2), for (a) Exp. 1: isoprene + NO₃, (b) Exp. 3: α-pinene + NO₃, and (c) Exp. 4: β-caryophyllene + NO₃.**





The comparison between the two filters during dark aging (Pre 2 – Pre 1 in Fig. 3) shows relatively small changes of the
chemical composition that are similar for all three systems: the mass fractions of high molecular weight compounds increase
and the mass fractions of low molecular weight compounds decrease. The signal fraction changes of individual compounds
are less than 2 %. Such change is mainly due to evaporation of semi-volatile compounds, as gas-phase wall losses act as a sink
for semi-volatile compounds, resulting in the repartitioning of the particle phase (Bertrand et al., 2018). Bell et al. (2021) show
in the α-pinene SOA system that most of the changes in chemical composition occur over the first 2 h of dark aging, meaning
at the time directly prior to photolysis the chemical composition has mostly stabilized.

During photolysis, the changes observed in the mass spectral patterns (Post – Pre 2 in Fig. 3) differ from those during dark
aging, and have a larger magnitude. For the isoprene system, the compounds exhibiting the largest decays in signal fraction
are $C_{10}H_{17}N_3O_{12-13}$ and $C_{10}H_{16}N_4O_{14-15}$ (8 % and 2 % of the total signal, respectively). At the same time, there are a few
compounds, e.g. $C_9H_{14}N_2O_7$ and $C_{10}H_{14}N_2O_9$, whose signal fractions increase (< 1 %). For the α-pinene system, the main dimer
compounds ($C_{20}H_{32}N_2O_{8,9,10}$) decrease by 13.6 % of the total signal during photolysis, while a few monomers, e.g. $C_9H_{15}NO_{6-}$
$_8$, $C_{10}H_{15}NO_{8-9}$ increase by up to 2 %. The β-caryophyllene SOA system shows decreases of signal fractions of both monomers
and dimers, mainly $C_{15}H_{25}NO_8$, $C_{15}H_{23}NO_9$ and $C_{15}H_{26}N_2O_9$, and $C_{30}H_{48}N_2O_{11-13}$, while the mass fractions of quite a few
compounds including $C_{12-14}$ compounds (e.g. $C_{14}H_{23}NO_{6,7}$), $C_{15}$ compounds (e.g. $C_{15}H_{23}NO_6$, $C_{15}H_{25}N_2O_8$), $C_{16-29}$ compounds
(e.g. $C_{29}H_{46}N_2O_{9-11}$) and $C_{30}$ compounds (e.g. $C_{30}H_{46}N_2O_{12}$) increase. Overall, our results show that the major compounds
decomposing during photolysis are both monomers and oligomers and that there are also compounds formed that contain fewer
carbon numbers and/or fewer nitrate groups.

The FIGAERO-CIMS offers insight into the pre- vs. post-photolysis SOA composition, and the EESI-TOF allows monitoring
the evolution of individual compounds at high time resolution. Using a linear fit to the 2 h pre-photolysis time series of the
individual compounds measured by the EESI-TOF and comparing a predicted value based on this fit at 1h post photolysis
(with 95 % confidence interval) with the actually measured signal (see the SI, Fig. S6), we find that 54 (the average value from
Exp. 1 and Exp. 2) out of 359, 24 out of 158 and 104 out of 441 compounds decreased significantly during photolysis, i.e., by
44 % ± 20 %, 64 % ± 24 %, and 24 % ± 18 %, for the isoprene, α-pinene and β-caryophyllene SOA, respectively. These
compounds contributed 53 %, 45 % and 62 % to the total pre-photolysis mass for these three systems. Meanwhile, there were
116, 53, and 164 compounds that increased during photolysis for the isoprene, α-pinene and β-caryophyllene SOA,
respectively. We note, however, that the fraction appearing to be resistant to photolysis or even showing increase may undergo
functional group changes during photodegradation even if the molecular formula appears unchanged during exposure to UV
light, or that decay of molecules with a specific molecular formula may be compensated by production of molecules with this
formula through the decay of other molecules. Here, we cannot fully exclude the particle-phase compounds volatilization
resulting from an increase in chamber temperature (3–4 °C, caused by heating from the chamber lights). However, because
most of the compounds showing decays are oligomers with very low predicted volatilities, such an increase in temperature is
unlikely to play a major role in the change in/of composition.



Figure 4 shows the time series of the main compounds that degraded during photolysis for each system, measured by both the EESI-TOF and the FIGAERO-CIMS. Within the error tolerances, the time series of the compounds measured by the two
instruments have a good agreement. The decay rates of these compounds are compound-dependent. Even though 1 h of photolytic aging might be too short to observe full decay trajectories (Zawadowicz et al., 2020; O'Brien and Kroll, 2019), it is long enough to distinguish the non-photolabile compounds and the photolabile compounds, and compare the decay rates of the photolabile compounds (Henry and Donahue, 2012; Wong et al., 2015). As also shown by Nah et al. (2016), most of the changes of α-pinene + $NO_3$ SOA due to photochemical aging happened in the first 1 h of light exposure.

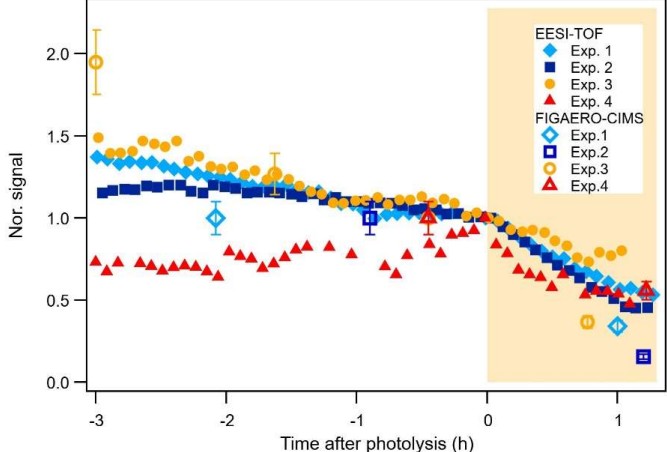


**Fig. 4 Time series of the isoprene tracers $C_{10}H_{17}N_3O_{12,13}$ (Exp.1 and Exp. 2), the α-pinene tracers $C_{20}H_{32}N_2O_{8-10}$ (Exp. 3) and the β-caryophyllene tracers $C_{30}H_{48}N_2O_{8-10}$ (Exp. 4) from the EESI-TOF and the FIGAERO-CIMS.**

Photolysis requires chromophores, which absorb particular wavelengths of visible light. Earlier studies of photodegradation of OH- and $O_3$-initiated SOA under low-$NO_x$ conditions have demonstrated that photolysis fragments molecules with functional
groups like carbonyls (C=O) and peroxides (R-O-O-R) (Surratt et al., 2006; Walser et al., 2007; Mang et al., 2008; Pan et al., 2009; Krapf et al., 2016). In contrast to these studies, the $BSOA_{NO3}$ in our study is rich in nitrate groups and oligomers. In Fig. 5a, we plot the average number of nitrate groups per monomer (monomer unit) for the three FIGAERO-CIMS filter samples (the same as in Fig. 3) selected before and after photolysis for all three SOA systems. Overall, the particle-phase compounds of all three SOA types contain on average a similar number of nitrate groups per monomer (1–1.5). During the last 2–3 h of
dark aging, the nitrate group fraction is stable, but photolysis causes a slight decrease of the nitrate-to-monomer ratio for all systems, consistent with the decrease of 1–3 % of the mass fraction of ONs of total organic compounds for all systems. It is clear from this that UV light fragmentation of nitrate groups only plays a minor role in changing the chemical composition of SOA when transitioning from dark to light conditions. It is known that organic nitrates can undergo photolysis and release $NO_2$ via $RONO_2 + hv \rightarrow RO\cdot + NO_2$ (Barnes et al., 1993). Measurements of the absorption cross-sections of a number of alkyl
nitrates, such as methyl nitrate, and some bifunctional organic nitrates have shown lifetimes up to between 14 h and 6 days

(Barnes et al., 1993). The absorption cross-sections of organic nitrates normally have an absorption maximum at $\lambda < 290$ nm and start to decrease when $\lambda > 310$ nm (Barnes et al., 1993; He et al., 2021). The backlights in the chamber reproduce the solar spectrum well in the range of 320–400 nm but the intensity at lower wavelengths (< 320 nm) falls off faster than the solar spectrum. Thus, the photolysis of nitrate groups observed in our study represents a lower limit of that would be expected in the atmosphere. More recent studies have shown that the presence of a nitrate group can enhance the absorption cross sections and make the photolysis of carbonyl nitrates faster than their reaction with OH (Müller et al., 2014; Suarez-Bertoa et al., 2012). However, in our study and on the time scales of our experiments, the cleavage of the nitrate functional group from the carbon chain is not the main loss pathway.

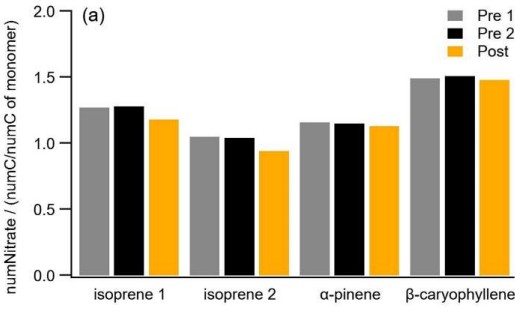

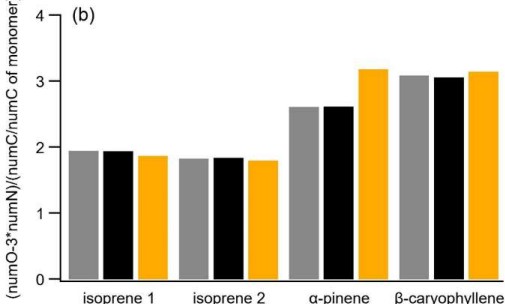

**Fig. 5 Mass-weighted number of nitrate groups to monomer (number of carbon/number of the carbon atoms in precursors (5, 10 and 15 for the isoprene, α-pinene and β-caryophyllene SOA, respectively)) ratio and mass-weighted number of non-nitrate oxygen to monomer ratio for three filters (Pre 1, Pre 2 and Post) for all experiments.**

In terms of oligomers, for the α-pinene and β-caryophyllene systems, the major degrading compounds are $C_{20}$ and $C_{30}$ dimers, while the major compounds formed are their corresponding $C_{10}$ and $C_{15}$ monomers (the isoprene system is a mix of $C_{10}$ dimers, $C_{15}$ trimers and $C_{20}$ tetramers, which makes it challenging to identify the potential parent compounds for those newly formed compounds). This indicates that for some fraction of the oligomers, the linkage between two monomer blocks is photolabile. Oligomers may be formed through both gas- and particle-phase processes. As $RO_2 + R'O_2 \rightarrow ROOR' + O_2$ is the main reaction





channel for RO₂ self- and cross-reactions in the gas phase, a large fraction of peroxides can be expected, and such mechanisms of dimer formation of the isoprene + NO₃ SOA were proposed (Ng et al., 2008; Zhao et al., 2020; Wu et al., 2020). Recent

studies also showed multiple pathways of dimerization in the particle phase or the gas-particle interface (Zhao et al., 2019; Claflin and Ziemann, 2018). All the studies mentioned above show that formation of dimers, in both the gas- and particle-phases, changes the functional groups of the resulting molecules compared to their building blocks (monomers). It is likely that peroxide functional groups are formed during RO₂ self- / cross-reactions, which could make some dimers photolabile, while the formation of potentially photolabile carbonyl functional groups is less certain from NO₃ oxidation.

Some of the compounds being formed during photolytic aging are molecules with 1-3 fewer carbon atoms than their potential parent compounds (e.g. $C_{13}$, $C_{14}$, $C_{29}$ compounds in the β-caryophyllene system), indicating fragmentation of the carbon skeleton. On average, each monomer (or monomer unit) of the isoprene, α-pinene and β-caryophyllene SOA contains ca. 2, 2.5, and 3 oxygen atoms pre photolysis (disregarding the oxygen atoms in the nitrate groups), respectively (Fig. 5b), suggesting that in addition to nitrate groups, these compounds also possess other oxygenated functional groups, and other possible

chromophores. The β-caryophyllene SOA exhibited higher losses of the particle mass and reduction in size (Fig. 1), followed by the α-pinene SOA, which could partly be explained by its higher number of functional groups per monomer (or monomer unit), i.e. larger photolabile fraction, compared to the other two systems.

Overall, the particle mass loss of nitrate-initiated SOA after 1h photolysis in our study is less than that reported for the OH- and O₃-initiated SOA under low-NOₓ conditions (Zawadowicz et al., 2020; Wong et al., 2015; Henry and Donahue, 2012;

Epstein et al., 2014), despite observing larger changes in the chemical composition. One possible reason could be that our systems have a large fraction of oligomers and photolysis-derived products may not directly evaporate, meaning a substantial fraction of these products retain low enough volatility to remain in the particle phase. In addition, in terms of different functional groups, addition of a nitrate group in a given molecule is estimated to reduce its vapour pressure by about 2.5 orders of magnitude, which is comparable to other oxygenated functional groups often observed in the OH- and O₃-initiated BSOA,

e.g. the hydroxyl group (−OH) and hydroperoxy group (−OOH) (2.4–2.5 orders of magnitude), or the carbonyl group (=O) (1 order of magnitude) (Pankow and Asher, 2008; Donahue et al., 2011). Thus, the gas-phase compounds formed in our systems could contain less functional groups (disregarding nitrate groups) than those in OH- and O₃-initiated BSOA, but still be of low enough volatility to condense into the particle phase. The elemental oxygen-to-carbon (O:C) ratio (disregarding the oxygen atoms in the nitrate groups) of the α-pinene SOA was calculated to be 0.26 ± 0.01 before photolysis in Exp. 3, which is lower

than the reported O:C ratios of 0.34–0.36 for BSOA from ozonolysis of α-pinene measured by the FIGAERO-CIMS in laboratory experiments (Huang et al., 2018). As shown above, nitrate groups might be less photolabile than carbonyls and peroxides, thus different sensitivities towards photolysis can be expected for the systems initiated by different oxidants.



### 3.4 Changes in the gas phase during photolysis

The primary focus of this study is on the changes in condensed-phase chemical composition of $BSOA_{NO3}$ due to photolytic aging. However, we cannot decouple processes happening in both particle and gas phases. For example, photo-degradation of semi-volatile compounds in the gas phase could lead to a decrease of compounds in the particle phase as they will evaporate to re-establish equilibrium. If gas-phase photo-degradation were the dominant cause for mass loss in all systems, then the largest decays would be expected from the most volatile species. Further, large non-volatile molecules (e.g. dimers in the β-

caryophyllene SOA) should be non-responsive to such a pathway. Because there is a systematic degradation of dimers in all systems, it is unlikely that repartitioning is driving the change in SOA composition during UV aging.

For the experiments shown here, the FIGAERO-CIMS using $I^-$ as the reagent ion (and an x-ray generator as the ion source) also measured the gas phase. However, in systems with a high abundance of $NO_3$ radicals, charge transfer can cause losses of $I^-$ and formation of $NO_3^-$ (Lee et al., 2014). Additionally, the reaction of $N_2O_5$ and $I^-$ may partly result in $NO_3^-$ + $INO_2$, and H

transfer reactions can occur between $I^-$ and $HNO_3$ ($HNO_3$ can come from the hydrolysis of $N_2O_5$ in our injection inlet or on chamber walls), both providing other routes to the formation of $NO_3^-$. As the amount of $NO_3^-$ introduced in our systems was very high, the signal of $NO_3^-$ was comparable to and sometimes even higher than the signal of $I^-$. Thus, gas-phase compounds were detected clustered with both $NO_3^-$ and $I^-$, with changing ratios of $I^-/NO_3^-$ during the experiments (higher during dark aging and lower during photolysis). This was not an issue for the particle phase, as particles are desorbed from the filter with pure

nitrogen. We probed the uncertainties of detected signals due to the changing $I^-/NO_3^-$ ratio (from ~0.1 to 1.6) after Exp. 2 (details see the description in the SI and Fig. S3). In our system, for those compounds clustered with $I^-$, the normalized signal didn't change significantly with decreasing $I^-/NO_3^-$ as long as $I^-/NO_3^-$ was higher than ~0.3, but it decreased by about 20–40 % when $I^-/NO_3^-$ decreased further to 0.1, as more molecules were clustered with $NO_3^-$.

Taking these uncertainties into account, we found that for the β-caryophyllene SOA, the signal of the major gas-phase

compounds either stayed stable (e.g. $C_{15}H_{25}NO_{5,6,7}$) or increased significantly (e.g. $C_{14}H_{23}NO_4$, $C_{15}H_{23}NO_6$, $C_{15}H_{26}N_2O_8$, with increases of the normalized signals of more than 5 times) during photolytic aging. For the isoprene and α-pinene SOA, although the low $I^-/NO3^-$ ratio (< 0.1) hindered a quantitative comparison, we still observed increases of several major compounds (e.g. $C_5H_{10}N_2O_8$ for the isoprene SOA, $C_{10}H_{15}NO_8$, $C_{10}H_{17}NO_7$ for the α-pinene SOA). These are likely fragmentation products from oligomers and/or monomers. As the volatility of these small compounds is much higher than their parent oligomer

compounds, they would tend to entirely or partly (depending on their volatility) evaporate to the gas phase. Different from the photolytic aging of $O_3$- or OH- initiated SOA, we did not however observe a significant increase of the signals of small acids in the gas phase, such as formic acid and acetic acid (Pan et al., 2009), despite observations of fragmentation reactions in the particle phase. Their minor importance seems not to significantly impact the measured formic or acetic acid in the gas phase.

In addition to photolytic aging of gas-phase compounds, there are other gas-phase reactions initiated by UV irradiation, which

may impact the chemical composition, e.g. OH oxidation, and the reactions with $O_3$ or $NO_3$ radicals. In our experimental conditions (200+ ppb $NO_2$ from decomposition of $N_2O_5$), if there are any OH radicals formed, then they will nearly exclusively




react with $NO_2$ given the high concentrations. For the reactions with $O_3$ or $NO_3$, if there were appreciable such reactions in the experiments with isoprene and β-caryophyllene, where double bonds are still present, we would expect to observe significant formation of more highly oxygenated molecules, which are absent in Fig. 3. Additionally, Fig. 3 shows there is a consistent

degradation of dimers in all systems, including α-pinene where double bonds are not retained, thus suggesting the dominant photolytic degradation pathways are independent of gas-phase chemistry.

**3.5 Impacts of photolysis on particle volatility**

Considering photolytic aging results in the degradation of molecules, there is a shift of the volatility distribution towards higher volatility species, which can cause evaporation of particulate mass. Here we discuss how the apparent volatility of remaining

particles changes during photolytic aging, and the potential influence the chemical processes discussed previously have on these changes.

Figure 6 shows the changes in the bulk volatility derived from different methods (bulk $\log_{10}C^*$ derived from molecular composition using different parametrizations, bulk $\log_{10}C^*$ derived from the kinetic model using the VTDMA measurements, and mass-weighted average $T_{max}$ from the FIGAERO-CIMS thermograms) during dark aging and photolysis for all $BSOA_{NO3}$

systems. Note that we show the $\Delta \log_{10}C^*$ (calculated by subtracting the bulk $\log_{10}C^*$ of the filter Pre 1 from the bulk $\log_{10}C^*$ of each filter), since the variabilities in the absolute bulk $\log_{10}C^*$ calculated by different parameterizations span several orders of magnitude: the bulk $\log_{10}C^*$ for all three systems is in the range of -11.9–-7.6 based on the parameterization by Mohr et al. (2019), -3.7–-0.3 by Li et al. (2016) and Isaacman-VanWertz and Aumont (2020), and -0.5–0 by Peräkylä et al. (2020) (see the SI, Fig. S7). Whereas a detailed discussion on these discrepancies is beyond the scope of this paper, they clearly reflect the

uncertainties related to volatility estimates of complex organic particles (O'Meara et al., 2014; Wu et al., 2020). $T_{max}$ is a qualitative measure of volatility, and is influenced by filter mass loadings, temperature ramp rate, and FIGAERO geometry (Huang et al., 2018; Schobesberger et al., 2018; Thornton et al., 2020). However, $T_{max}$ variation due to these factors is generally smaller than the difference in monomers and dimers. We compared the shape of thermograms and $T_{max}$ of several monomers and dimers observed in our study to that of the same compounds from a few field campaigns (see the SI, fig. S8). Overall, the

thermal desorption behaviour observed in our chamber experiments is similar compared to this from field measurements. By further comparing the $\log_{10}C^*$ of individual compounds based on e.g. Mohr et al. (2019) with their corresponding $T_{max}$ (Fig. S9), we found a generally good qualitative agreement, but also noticed large uncertainties in the calculated $\log_{10}C^*$, especially for ON oligomers with multiple nitrate groups. For example, the dominating dimer $C_{20}H_{32}N_2O_9$ has a $T_{max}$ of about 70 °C, which is in the range of $T_{max}$ of $C_{10}$ monomers such as $C_{10}H_{15}NO_9$. However, based on the parameterization of Mohr et al.

(2019), the $\log_{10}C^*$ of the dimer $C_{20}H_{32}N_2O_9$ (-7.9) is much lower than the $\log_{10}C^*$ of the $C_{10}$ monomers ($\log_{10}C^* > -5$).



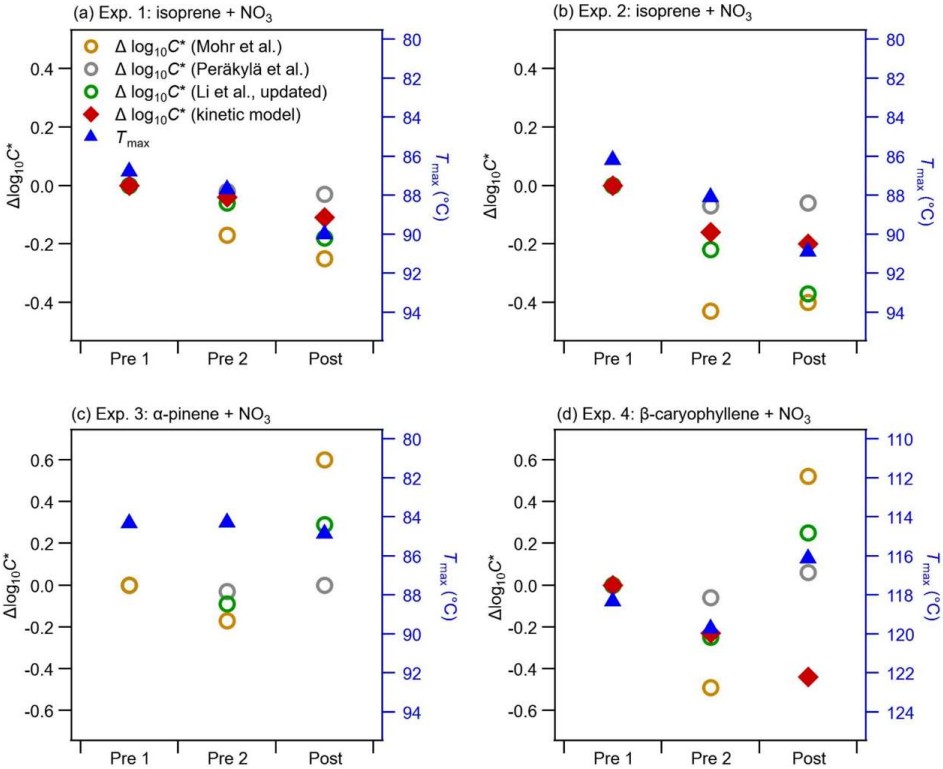

**Fig. 6 Comparison of changes in the bulk volatility measured with three methods before and after photolysis (Pre 1, Pre 2 and Post) for all experiments. Method 1: $\Delta\log_{10}C^*$ based on the parameterizations of Mohr et al. (2019), Li et al. (2016) (updated by Isaacman-VanWertz and Aumont (2020)), and Peräkylä et al. (2020); Method 2: mass-weighted average $T_{max}$ (note the right axes are swapped, and the range in (d) is different from that in others); Method 3: $\Delta\log_{10}C^*$ from the kinetic model.**

Overall, for the two isoprene experiments (Exp. 1 and Exp. 2), almost all methods show a decrease in volatility as the experiment progresses, with no change in trend during the transition from dark to light conditions. This agrees well with the changes in the chemical composition as shown in Fig. 3: during dark aging, the chemical composition shifts to higher mass (lower volatile) compounds and causes a decrease in the bulk volatility; during photolysis, there is additional decay of some oligomers such as $C_{10}H_{17}N_3O_{13}$, whose volatility is in the intermediate range of all compounds (Fig. S9). As a consequence, despite clear changes in the chemical composition, the bulk volatility does not change substantially due to photolysis. In Exp. 2, the $\Delta\log_{10}C^*$ by both Peräkylä et al. (2020) and Mohr et al. (2019) decreased during the dark aging but increased slightly during photolysis, which is different compared to other methods. The difference between Exp. 1 and Exp. 2 is probably due to the differences in the chemical composition (Fig. 2 vs. Fig. S5) as Exp. 1 had higher $NO_3$/isoprene ratio and the SOA with





more nitrate groups (Fig. 5), and also due to different mechanisms to treat nitrate groups in the different parameterizations. It reflects again uncertainties related to volatility estimates by different parameterizations.

In Exp. 3 (α-pinene + NO$_3$), no clear trend in volatility can be discerned. $T_{max}$ does not exhibit significant changes during both dark aging and photolysis. However, the $\Delta\log_{10}C^*$ by the parameterizations shows a slight decrease during the dark aging and an increase during photolysis. As shown in Fig. 3, we observed decreases of a few dimers and increases of a few monomers,
which, based on the parameterizations, causes a shift from low volatile compounds to high volatile compounds. However, as mentioned above, the $T_{max}$ of these dimers is not necessarily higher than those monomers, i.e., the volatility of these dimers is not necessarily lower than that of those monomers. Thus, the mass-weighted average $T_{max}$ doesn't decrease. Unfortunately, no VTDMA data are available for this experiment.

In Exp. 4 (β-caryophyllene + NO$_3$), the different methods yield different information on the relative changes of volatility from
dark to photolytic aging. Both $T_{max}$ and $\Delta\log_{10}C^*$ from the parameterizations show the bulk volatility decrease during dark aging and increase during photolysis, which would agree with changes in chemical composition. The shift from small (high volatile) compounds to large (low volatile) compounds during dark aging causes the decrease in the bulk volatility, while during photolysis the major compounds that decay cover both monomers and oligomers with a wide range of volatility, and there are a few newly formed C$_{<15}$ compounds that remain in the particle phase (Fig. 3), thus the bulk volatility increases.
However, the $\Delta\log_{10}C^*$ from the VTDMA data agrees with the trend during dark aging but not during photolysis. We note that in Exp. 1–3, the particle size distribution and mean particle size, as shown in Fig. 1, did not change substantially, while in Exp. 4 (β-caryophyllene + NO$_3$), the particles size became significantly smaller during photolysis (Fig. 1). In Fig. 7, the evolution of the volume distribution of the β-caryophyllene SOA in Exp. 4 shows the size distribution slowly shifts to larger sizes during dark aging, but during photolysis the decrease in particles with diameters larger than 300 nm is much greater than the decrease
in small particles, indicating that the photo-degradation is dependent upon particle size. The size dependence can be explained by calculating the mass absorption efficiency as a function of particle size (see the SI, Fig. S4), which shows particles with a diameter of 300 nm are 1.5–1.7 times more efficient in absorbing the chamber lights ($\lambda = 350$ nm) than 100 nm particles. Other processes, such as coagulation and evaporation, tend to lead to larger decreases of smaller particles, which is opposite to what we found here, while the wall loss rate (which is tested in another experiment during the same campaign) was nearly constant
for the particle size range up to 400 nm, thus unlikely the reason for size dependence of light absorption. In comparison, the isoprene and α-pinene systems have smaller and narrower size distribution, and consequently do not have an equally important size-dependence in their response to UV light. This could also partly explain why the β-caryophyllene SOA had the largest decay in particle mass and size among the three systems. As the particles with different sizes will not have a uniform change in their chemical composition, the volatility measured for one selected size by the VTDMA may not be able to represent the
change in the bulk volatility for experiments with a wide distribution of particle sizes–similarly, caution should be applied when interpreting the representativeness of measurements based on the bulk mass for the whole particle size distribution.





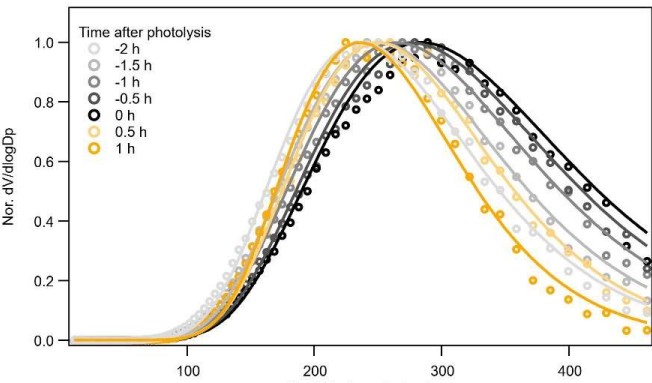

**Fig. 7 Evolution of particle size distribution of the β-caryophyllene SOA during dark aging and photolysis. The solid lines are lognormal fittings.**

## 4. Conclusion & Atmospheric Implications

Our experimental observations on three BSOA$_{NO3}$ systems show that the dark-to-light transition (1 h photolytic aging) causes slight to moderate evaporation in BSOA$_{NO3}$ (~0, 3.5 and 12 % of the total mass for the isoprene, α-pinene and β-caryophyllene SOA, respectively). Evaporation is due to the fragmentation of photolabile compounds and formation of volatile fragmentation products. Despite finding 0–12 % of the mass evaporating during photolysis, 53 %, 45 % and 62 % of the total mass of the isoprene, α-pinene, and β-caryophyllene SOA, respectively, is sensitive to photolytic aging, representing a majority of the pre-photolysis composition.

Fragmentation of nitrate groups is not the main loss pathway on the time scale of our experiments. For all BSOA$_{NO3}$ studied here, UV light fragments oligomers at the linkage between the monomer units (likely peroxides), as well as functional groups at other positions, causing the formation of monomers and compounds with shorter carbon skeleton (e.g. $C_{10} \rightarrow C_9$). The newly formed compounds are more volatile than their parent compounds, and are thus entirely/partly released back into the gas phase. The changes in bulk volatility vary with individual experiments, and depend on (1) whether the volatility of photolabile fractions is higher/lower than that of non-photolabile fractions (2) how the fragment products remain in the particle phase. The comparisons of different methods to assess the bulk volatility reveal the complexity of volatility assessments. There is generally a good agreement with the trends between the VTDMA and thermal desorption behaviour from the FIGAERO-CIMS (both desorption-based methods). However, the uncertainties related to molecular formula-derived parameterization, especially for the compounds with multiple nitrate groups, likely causes disagreement when assessing the changes in the absolute bulk volatility. It is also worth noting that particle size dependence of light absorption should be considered when studying the changes in the bulk volatility for systems with broad size distributions.



Our systems represent conditions in which $RO_2$-$RO_2$ and $RO_2$-$NO_3$ reactions are favoured and ON oligomers contribute to the
high SOA yields we observed. Different changes in particle mass/size due to photolysis could be expected for the $BSOA_{NO3}$
formed under different conditions, as they have different chemical composition, thus different amounts and/or types of
chromophores. In addition, the differences among the three BVOC systems also indicate that results obtained for specific
BVOC cannot directly be generalized to SOA present in the atmosphere and it is necessary to study different BVOC-SOA
systems.

To our knowledge, the photolysis of SOA has so far been studied mainly under low-NOx conditions. The sensitivity of ONs
to UV light is poorly understood. As ONs are important in high-$NO_x$ environments and make substantial contributions to the
total organic aerosols (Farmer et al., 2010; Lee et al., 2016), future studies that probe the photolytic aging of ONs under
different ambient conditions and for different SOA precursors are needed to improve our understanding of the life cycle of
ONs.

**Data availability**

The datasets are available upon request to the corresponding authors.

**Author contributions**

CW, DMB and CM designed the study. Chamber experiments were carried out by CW, DMB, ELG, SH, AB, SG and CM.
Data analysis and interpretation was performed by CW, DMB, ELG, IR, JS and CM. CW wrote the manuscript, with input
from all co-authors. All co-authors read and commented on the manuscript.

**Competing interests**

The authors declare that they have no conflict of interest.

**Acknowledgements**

This work was supported by the EUROCHAMP-2020 and the Knut and Alice Wallenberg Foundation (WAF project
CLOUDFORM, grant no. 2017.0165).





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
