# Peer review of "Photolytically Induced Changes in Composition and Volatility of Biogenic Secondary Organic Aerosol from Nitrate Radical Oxidation during Night-to-day Transition"

_Atmospheric Chemistry and Physics, 2021_

## Referee Comment (RC1)

Overall comment:

This work examined the composition and volatility changes of SOA formed from NO3 oxidation of three biogenic VOCs upon UV photolysis. This study used two state-of-the-art SOA molecular analysis instruments, FIGAERO-CIMS and EESI-TOF to examine the SOA composition change before and after photolysis. The authors also compared a few methods that were used to estimate SOA bulk volatility. The SOA compositions were shown to be very interesting that the oligomer fractions were extremely high (higher than ever reported for SOA systems) for all the three biogenic VOCs. The comparisons of bulk volatility estimates lead to a conclusion that SOA that largely contain nitrate groups are highly uncertain. Overall, the manuscript was well written, and the experiments and analyses were carefully carried out. However, there are some major concerns regarding the interpretation of the instruments and chemistry in the NO3 oxidation. These concerns should be addressed before the manuscript can be published in ACP.

Major comments:

1. Line 134 – 139. The determination of negligible fragmentation fractions due to thermal desorption is unclear. For a certain m/z or fitted chemical formula, FIGAERO-CIMS can usually produce an overall thermogram with a broad peak or peaks, each of which can be integrated by a number of individual species' thermogram peaks (Schobesberger et al., 2018, ACP). Deconvoluting the individual peaks might involve large uncertainties. When there is more than one broad peaks, it is apparent that the later broad peak is from thermal decomposition of a much lower volatility compound such as dimers (e.g., Figure S2a). But thermal decomposition peaks can also be hidden under the same broad peak, especially when the parent compound and the fragmentation compound have more close volatilities. For example, assuming a compound with formula $C_xH_yO_zN$ undergo thermal decomposition (dehydration) and produces $C_xH_{y-2}O_{z-1}N$, the decomposition product could be difficult to separate from original $C_xH_{y-2}O_{z-1}N$ in the broad thermogram peak. The two steps described in the supporting information may not be able to identify such thermal decomposition. Therefore, the percentage ranges provided in Line 134 – 135 are likely the lower estimate of the thermally labile species in these SOA.

2. This is a related questions regarding thermal decomposition. In NO3 + biogenic VOC oxidation systems, it is likely common to have PAN species from RO2 + NO2 (NO2 as a byproduct from N2O5 decomposition). PAN species are very thermally labile and produce RO2 + NO2. This may happen even under moderate heating temperatures. Do the authors have a clue to what extent this chemistry has happened in FIGAERO-CIMS? This class of species was not mentioned throughout the manuscript.

3. Line 177. The authors clarified how particle mass concentration was corrected for wall loss and coagulation. But it is unclear how the ion signals from FIGAERO-CIMS and EESI-TOF were corrected, especially for coagulation. Can the detail be elaborated?

4. The reported oligomer fractions are very high in this work. In fact, I have not seen any studies reporting such high fractions of oligomers and wonder what could have caused these results. For the a-pinene + NO3 system, Nah et al. (2016 ES&T) found that the majority of the SOA was made by monomers. The discrepancy could be due to two different conditions:

(i) Nah et al. used much lower VOC (~ 10 ppb) vs. 100 ppb in this work; and
(ii) Nah et al. had RO2 + HO2 as the dominant condition vs. RO2+RO2 in this work;
But even these differences are responsible for the large oligomer fractions reported in this work, how is it possible that monomer products from RO2 + RO2 did not partition and form SOA? The SOA mass loadings were still > 10 ug/m3. Some discussion or explanation should be provided.

5. Line 304 and 310. Can you elaborate why CHO compounds were more prominent in EESI-TOF and what "degradation pathways" were referred to? In principle, the FIGAERO-CIMS requires heating and should be more likely to decompose CHON compounds than the EESI-TOF.

6. Can the authors show the changes of particle-phase N2O5 and HNO3 signals (FIGAERO-CIMS) under the various conditions? Since N2O5 is likely in excess and the experiments were carried out under humid conditions, it is interesting to see how N2O5 uptake might have affected the chemistry. In fact, the chemical compositions shown in Figure 2 have many peaks with 3-4 N in the formulas. Could they have to do with particle-phase chemistry related to N2O5 uptake, rather than just gas-phase oxidation?

7. SOA volatility. The authors compared a number of methods to estimate SOA volatilities and the results are very different. The authors concluded that the molecular formula derived parameterization is highly uncertain, especially for the compounds with multiple nitrate groups. Can the authors break down the comparison by #N? For example, how different are the methods for species with 0-4 nitrogen atoms? Also, dimers with multiple N atoms were found to have similar Tmax as monomers. Could this be real or some unrecognized artifact in FIGAERO-CIMS? If this were true, it somewhat contradicts the earlier discussions regarding evaporation of SOA after photolysis.

Minor comments:

1. Line 91. Change "outside" to "out".

2. Line 407 – 408. Some evidence to support this statement should be added.

---

## Author Comment (AC1)

Response Letter to Referee #1

The authors thank the reviewer for the careful review of our manuscript and helpful comments and suggestions. All the comments (in bold text) are addressed below point by point, with our response following in non-bold text and the corresponding revisions to the manuscript in blue. All updates of the original submission are tracked in the revised version.

**Overall comment:**

**This work examined the composition and volatility changes of SOA formed from $NO_3$ oxidation of three biogenic VOCs upon UV photolysis. This study used two state-of-the-art SOA molecular analysis instruments, FIGAERO-CIMS and EESI-TOF to examine the SOA composition change before and after photolysis. The authors also compared a few methods that were used to estimate SOA bulk volatility. The SOA compositions were shown to be very interesting that the oligomer fractions were extremely high (higher than ever reported for SOA systems) for all the three biogenic VOCs. The comparisons of bulk volatility estimates lead to a conclusion that SOA that largely contain nitrate groups are highly uncertain. Overall, the manuscript was well written, and the experiments and analyses were carefully carried out. However, there are some major concerns regarding the interpretation of the instruments and chemistry in the NO3 oxidation. These concerns should be addressed before the manuscript can be published in ACP.**

We thank the reviewer for the positive assessment of our manuscript.

**Major comments:**

**1. Line 134 – 139. The determination of negligible fragmentation fractions due to thermal desorption is unclear. For a certain m/z or fitted chemical formula, FIGAERO-CIMS can usually produce an overall thermogram with a broad peak or peaks, each of which can be integrated by a number of individual species' thermogram peaks (Schobesberger et al., 2018, ACP). Deconvoluting the individual peaks might involve large uncertainties. When there is more than one broad peaks, it is apparent that the later broad peak is from thermal decomposition of a much lower volatility compound such as dimers (e.g., Figure S2a). But thermal decomposition peaks can also be hidden under the same broad peak, especially when the parent compound and the fragmentation compound have more close volatilities. For example, assuming a compound with formula CxHyOzN undergo thermal decomposition (dehydration) and produces CxHy-2Oz-1N, the decomposition product could be difficult to separate from original CxHy-2Oz-1N in the broad thermogram peak. The two steps described in the supporting information may not be able to identify such thermal decomposition. Therefore, the percentage ranges provided in Line 134– 135 are likely the lower estimate of the thermally labile species in these SOA.**

We thank the reviewer for pointing out this important issue. However, what needs to be compared is the *thermal decomposition temperature* of compound $C_xH_yO_zN$ forming e.g. $C_xH_{y-2}O_{z-1}N$ (dehydration) or $C_{x-1}H_yO_{z-2}N$ (decarboxylation) with the *thermal desorption temperature* of compounds with the same formula as the thermal fragmentation products. Dehydration or decarboxylation reactions are expected to occur at temperatures higher than

120 °C (Buchholz et al., 2020; Stark et al., 2017), while the thermal desorption temperatures of major compounds in the isoprene and α-pinene systems and the monomers in the β-caryophyllene system are below 120 °C. As shown in Fig. S3 (which is added as a new figure to the supplementary information (SI) of our manuscript) below, most of the top 30 compounds in the isoprene and α-pinene systems and major monomers in the β-caryophyllene system do not have "shoulders" or "plateaus", and their signals in the temperature range higher than 120 °C are relatively low. Therefore, for these compounds, signals from desorbing compounds and thermal fragmentation products are well separated. There are a few exceptions with bi-modal thermograms (Fig. S3c, blue lines), which were corrected as described in the manuscript in the SI S1.2. Even though for some compounds with thermal desorption temperature higher than 120 °C (e.g. the dimers in the β-caryophyllene system, Fig. S3c&d), interference from fragmentation may be possible as mentioned by the reviewer, it is rather unlikely an issue in our study. The molecular formulae of the compounds with significant tailings (Fig. S3d) do not correspond to thermal decomposition products of major dimers (e.g. dehydration products with $m/z - 18.015$ compared to the parent compounds), but the tailings might rather be signal from isomers of these major dimers. Moreover, the major peaks observed by the FIGAERO-CIMS and the EESI-TOF, for which no thermal fragmentation can occur, are similar. For example, for the isoprene system, the dominating compound measured by both instruments is $C_{10}H_{17}N_3O_{13}$, and no substantial signal of $C_{10}H_{15}N_3O_{12}$ ($m/z -18.015$) is found.

We have added Fig. S3 to the SI of our manuscript, together with explanations based on the above response (S1.2): "Fig. S3 shows thermal fragmentation to be a minor issue in our study. Most of the peaks in the three systems do not have "shoulders" or "plateaus". In addition, for almost all compounds in the isoprene and α-pinene systems and the monomers in the β-caryophyllene system, the signals in the temperature range higher than 120 °C where dehydration or decarboxylation reactions are expected to occur (Buchholz et al., 2020; Stark et al., 2017) are relatively low. There are a few exceptions with bi-modal thermograms (Fig. S3c, blue lines), which were corrected for as described above. Regarding the dimers from the β-caryophyllene system (Figure S3d), there are a few compounds with significant tailings. However, their molecular formulae do not correspond to thermal decomposition products of major dimers (e.g. dehydration products with $m/z - 18.015$ compared to the parent compounds), but the tailings might rather be signal from isomers of these major dimers." Lines 135 – 141 in the revised manuscript were rewritten as following: "As shown in Fig. S3, most of the thermograms of individual compounds for all systems exhibited unimodal and sharp peaks. Thermal fragmentation products were detected either through thermograms of individual compounds with multiple peaks (normally double peaks) or one peak with $T_{max}$ (desorption temperature at which a compound´s signal exhibits a maximum) much higher than the estimated $T_{max}$ (e.g., for α-pinene SOA, a $C_{10}$ monomer had a $T_{max} \approx 140$ °C which is in the range of $T_{max}$ for $C_{20}$ dimers (Faxon et al., 2018)). Artefacts resulting from thermal fragmentation products were corrected for (see the SI S1.2 and Fig. S2)."

[Figure]

Fig. S3 Thermograms of top 30 (signal) compounds of the SOA from (a) Exp.1, (b) Exp. 3 and (c) Exp. 4 and (d) major dimers from Exp. 4.

**2. This is a related questions regarding thermal decomposition. In NO₃ + biogenic VOC oxidation systems, it is likely common to have PAN species from RO₂ + NO₂ (NO2 as a byproduct from N2O5 decomposition). PAN species are very thermally labile and produce RO2 + NO2. This may happen even under moderate heating temperatures. Do the authors have a clue to what extent this chemistry has happened in FIGAERO-CIMS? This class of species was not mentioned throughout the manuscript.**

We thank the reviewer for bringing up an interesting point. Since the information that can be obtained from mass spectrometers is molecular composition, but not molecular structure, we are limited in assigning the detected compounds to a certain group of compounds such as PANs. As the reviewer pointed out, PAN species could be formed in our system through $RO_2 + NO_2$, however, as oligomers are dominating in our isoprene and α-pinene systems, this termination pathway which produces monomer compounds is unlikely as important as the $RO_2 + RO_2$ pathway. For example in the α-pinene system, we observed monomers with molecular formulae corresponding to pinonaldehyde-PAN ($C_{10}H_{15}NO_6$) and norpinonaldehyde-PAN ($C_9H_{13}NO_6$) (Nah et al., 2016), but their signals were much lower than that of the dominating dimer $C_{20}H_{32}N_2O_9$ (Fig. 2). For the β-caryophyllene system, we did observe a big fraction of monomers and some of them have PAN-like molecular formulae, e.g. $C_{15}H_{23}NO_{7,8}$. However, overall, the sum of signals of $C_{15}H_yO_zN$ is 25 times higher than the sum of signals of $C_{15}H_yO_z$. Similarly, if the thermal decomposition of PAN species in the FIGAERO inlet was significant

for all three SOA systems, we would observe the thermal decomposition products as CHO (without nitrate groups). Since CHO compounds only contribute a very small fraction to the total signal, thermal decomposition of PAN species unlikely plays a significant role for the SOA generated in our study. In addition, assuming thermal decomposition would play a lesser role in the EESI-TOF compared to the FIGAERO-CIMS, more potential fragmentation products and fewer compounds with molecular formulae corresponding to PANs would be expected in the FIGAERO-CIMS mass spectra. By comparing the molecular compositions detected by the FIGAERO-CIMS and the EESI-TOF, we did not observe big differences in major compounds measured by two instruments.

**3. Line 177. The authors clarified how particle mass concentration was corrected for wall loss and coagulation. But it is unclear how the ion signals from FIGAERO-CIMS and EESI-TOF were corrected, especially for coagulation. Can the detail be elaborated?**

We apologize, this was not correctly stated in the manuscript. As the signal of the FIGAERO-CIMS and the EESI-TOF is based on particle mass and not particle number, no correction for coagulation is needed. Lines 178 – 184 were revised as following to clarify the correction of the FIGAERO-CIMS and EESI-TOF signal for wall loss (but not for coagulation): "They were corrected for wall losses using a uniform dynamic wall loss rate $k_{wall}$ for the whole size range. $k_{wall}$ was determined from the observed exponential decay of the particle number concentration (taking coagulation into account) using Eq. 2, where $N$ corresponds to the particle number concentration and $k_{coag}$ corresponds to the coagulation coefficient ($5*10^{-10}$ s$^{-1}$ (Pospisilova et al., 2020)).

$$\frac{dN}{dt} = -k_{coag} * N^2 - k_{wall} * N \qquad \text{(Eq. 2)}$$

The wall loss-corrected particle mass concentration was divided by the uncorrected mass concentration to obtain the wall loss correction factor applied to the EESI-TOF and FIGAERO-CIMS signal."

**4. The reported oligomer fractions are very high in this work. In fact, I have not seen any studies reporting such high fractions of oligomers and wonder what could have caused these results. For the a-pinene + NO3 system, Nah et al. (2016 ES&T) found that the majority of the SOA was made by monomers. The discrepancy could be due to two different conditions: (i) Nah et al. used much lower VOC (~ 10 ppb) vs. 100 ppb in this work; and (ii) Nah et al. had RO2 + HO2 as the dominant condition vs. RO2+RO2 in this work; But even these differences are responsible for the large oligomer fractions reported in this work, how is it possible that monomer products from RO2 + RO2 did not partition and form SOA? The SOA mass loadings were still > 10 ug/m3. Some discussion or explanation should be provided.**

For the isoprene+NO$_3$ system, it is not surprising that the oligomers are dominating, because the monomers are too volatile. For a typical C$_5$ compound with 2 nitrate groups, e.g. C$_5$H$_8$N$_2$O$_8$, which is the dominating gas-phase monomer observed in our study (and has been reported in other studies, e.g. Zhao et al. (2021)), the estimated $\log_{10}C^*$ at 300K is between 1.53 and 4.4

based on the three parameterizations used in this study (Mohr et al., 2019; Peräkylä et al., 2020; Isaacman-VanWertz and Aumont, 2021), i.e., in the SVOC range (Donahue et al., 2006).

For the α-pinene + $NO_3$ and β-caryophyllene + $NO_3$ systems, we did observe monomers (see Fig. 2 and also new SI figure Fig. S7), and the large contributions from oligomers can be explained by the dominating $RO_2$ + $RO_2$ and $RO_2$ + $NO_3$ pathways. In a more similar study to ours of α-pinene + $NO_3$ SOA, in which reactions between peroxy radicals ($RO_2$ + $RO_2$) and $RO_2$ + $NO_3$ are favoured (Takeuchi and Ng, 2019), the SOA measured by the FIGAERO-CIMS was also dominated by $C_{20}$ dimers. Their dominating dimer was $C_{20}H_{32}N_2O_9$ and the dominating monomer was $C_{10}H_{15}NO_6$, identical to our study. Their ratio of $C_{20}H_{32}N_2O_9$ to $C_{10}H_{15}NO_6$ was about 5, in our study ~ 2 times higher (about 10) (Fig. 2). Therefore, the large fraction of oligomers shown here are not so surprising, and likely come from the large concentrations of $RO_2$ radicals present.

In the revised manuscript, we have added this discussion after mentioning the study of Takeuchi and Ng (2019). Lines 302 – 307 were revised as following: "Our mass spectra are similar to those reported in Takeuchi and Ng (2019) measured with a FIGAERO-CIMS, where the α-pinene + $NO_3$ SOA was dominated by $C_{20}$ dimers with two dominating compounds, $C_{20}H_{32}N_2O_9$ and $C_{20}H_{32}N_2O_{10}$. The dominating monomer in their study was $C_{10}H_{15}NO_6$, also identical to our study. Their ratio of $C_{20}H_{32}N_2O_9$ to $C_{10}H_{15}NO_6$ was about 5, in our study ~ 2 times higher (about 10). In the system in Takeuchi and Ng (2019), reactions between peroxy radicals ($RO_2$ + $RO_2$) and $RO_2$ + $NO_3$ are also favoured. Large concentrations of $RO_2$ present in both studies likely lead to the large fraction of oligomers."

**5. Line 304 and 310. Can you elaborate why CHO compounds were more prominent in EESI-TOF and what "degradation pathways" were referred to? In principle, the FIGAERO-CIMS requires heating and should be more likely to decompose CHON compounds than the EESI-TOF.**

We are currently preparing a manuscript about the potential artefacts of the EESI-TOF. Briefly, in our $NO_3$-initiated systems with a high abundancy of dinitrate dimer species (or oligomers with an even higher number of nitrate groups), $C_xH_yO_z$ species that were observed by the EESI-TOF but not by the FIGAERO-CIMS such as e.g. $C_4H_6O_2$ (mentioned in line 292), $C_{10}H_{16}O_3$ and $C_{10}H_{16}O_4$ (mentioned in line 311) are likely related to artefacts from the electrospray ionization in the EESI-TOF (Goracci et al., 2017; James et al., 2006; Keith-Roach, 2010; Maire and Lange, 2010; Kourtchev et al., 2020; Rovelli et al., 2020). Details are mentioned in lines 328 – 330.

For clarification reasons we have rearranged the discussion on artefacts in the EESI-TOF in the revised manuscript as following: In lines 293 – 294, we deleted "to be discussed in a future paper (Bell et al. in preparation)". In lines 311 – 312, we deleted "which may also stem from some degradation pathways in the EESI-TOF source". Lines 324 – 330 were revised as following: "Differences are also caused by the fragmentation of oligomers into smaller compounds ($C_xH_yO_z$) inside the EESI-TOF, such as $C_4H_6O_2$ in the isoprene + $NO_3$ system, and a few CHO compounds in the α-pinene + $NO_3$ and β-caryophyllene + $NO_3$ systems. These compounds are likely related to artefacts from the electrospray ionization (Goracci et al., 2017; James et al., 2006; Keith-Roach, 2010; Maire and Lange, 2010; Kourtchev et al., 2020; Rovelli

et al., 2020). They may occur due to the proximity of $-ONO_2$ functional groups next to peroxy linkages which results in fragmentation when exposed to water in the electrospray (Bell et al, in preparation), and a loss of a $HNO_3$ fragment during ionization (Liu et al., 2019)."

**6. Can the authors show the changes of particle-phase $N_2O_5$ and $HNO_3$ signals (FIGAERO-CIMS) under the various conditions? Since $N_2O_5$ is likely in excess and the experiments were carried out under humid conditions, it is interesting to see how $N_2O_5$ uptake might have affected the chemistry. In fact, the chemical compositions shown in Figure 2 have many peaks with 3-4 N in the formulas. Could they have to do with particle-phase chemistry related to $N_2O_5$ uptake, rather than just gas-phase oxidation?**

For more details on the role of $N_2O_5$ during the dark aging period, we refer the reader to the companion paper by Bell et al. (2021), which is available online (https://acp.copernicus.org/preprints/acp-2021-379/). Briefly, with the FIGAERO-CIMS we observed a small fraction of $N_2O_5$ present in the particle phase, which could act as an oxidant. The fraction of $N_2O_5$ in the particle phase decreased with time during the experiment (by ~50% over ~3 h, shown in Figure S8 in Bell et al. (2021)), indicating it was either consumed or evaporating from the particle phase. Overall, $N_2O_5$ or peroxide-containing organics present in the particle phase could be responsible for the observed continued oxidation, most of which occurred over the first 2 h of dark aging, however.

For the particle-phase $HNO_3$, as the gas-phase $HNO_3$ signal was very high, the particle-phase signal was impacted by the gas-phase measurement. Thus, it is not possible to quantify the changes between the different samples.

**7. SOA volatility. The authors compared a number of methods to estimate SOA volatilities and the results are very different. The authors concluded that the molecular formula derived parameterization is highly uncertain, especially for the compounds with multiple nitrate groups. Can the authors break down the comparison by #N? For example, how different are the methods for species with 0-4 nitrogen atoms? Also, dimers with multiple N atoms were found to have similar Tmax as monomers. Could this be real or some unrecognized artifact in FIGAERO-CIMS? If this were true, it somewhat contradicts the earlier discussions regarding evaporation of SOA after photolysis.**

We thank the reviewer for these suggestions. In fact we have made a new figure where we show $\log_{10}C^*$ based on the parameterizations of Mohr et al. (2019), Li et al. (2016) (updated by Isaacman-VanWertz and Aumont (2021)), and Peräkylä et al. (2020) of major compounds with $1 - 4$ N from all three systems (Fig. S10 in the SI of the revised manuscript). In the small panel of Fig. S10, only $C_{10}$ compounds are compared in order to exclude the influence by different carbon numbers. It is apparent that the difference between the $\log_{10}C^*$ from the three parameterizations becomes larger with increasing nitrogen number. Similar comparison study performed by Wu et al. (2021) showed as well that the discrepancy in saturation vapor pressure from different parametrizations increases with increasing complexity of molecules. We have added the following sentence to lines $513 - 514$ of the revised manuscript: "As Fig. S10 in the SI shows, the discrepancy of $\log_{10}C^*$ from the three parameterizations becomes larger with increasing nitrogen number."

[Figure]

Fig. S10 $\log_{10}C^*$ based on the parameterizations of Mohr et al. (2019), Li et al. (2016) (updated by Isaacman-VanWertz and Aumont (2021)), and Peräkylä et al. (2020) of major compounds with 1 – 4 N from all three systems. In the small panel, only $C_{10}$ compounds are compared.

Concerning the comparison between $T_{max}$ of dimers and monomers, as is shown in an updated version of Fig. S9 (now Fig. S12) below, in general there is a clear trend of increasing $T_{max}$ with decreasing $\log_{10}C^*$, and $T_{max}$ is higher for dimers than corresponding monomers for all systems (the isoprene + $NO_3$ system was dominated by oligomers). In the β-caryophyllene + $NO_3$ system, dimers have a much higher $T_{max}$ (between 110 and 150 °C) than monomers ($T_{max}$ between 60 and 100 °C). Similarly, in the α-pinene + $NO_3$ system, most of the dimers have higher $T_{max}$ than that of monomers (Fig. S12, green circles), except for a few dimers, e.g. $C_{20}H_{32}N_2O_9$, which have similar $T_{max}$ as some monomers (e.g. $C_{10}H_{15}NO_9$). The possible reason could be that these dimers have less functional groups which lower their volatility, than those monomers (e.g. $C_{20}H_{32}N_2O_9$ has two nitrate groups, and three more oxygen atoms, while $C_{10}H_{15}NO_9$ has one nitrate group but six more oxygen atoms.).

In addition to replacing Fig. S9 with the updated version below, we modified lines 521 – 523 as following: "By further comparing the $\log_{10}C^*$ of individual compounds based on e.g. Mohr et al. (2019) with their corresponding $T_{max}$ (Fig. S12), we found good qualitative agreement with a clear trend of increasing $T_{max}$ with decreasing $\log_{10}C^*$, and higher $T_{max}$ for dimers than corresponding monomers for all systems."

[Figure]

Fig. S12 Average $T_{max}$ from all filter samples during dark aging and photolysis versus saturation concentration $\log_{10}C^{*}$ by Mohr et al. (2019) of the signal top 50 compounds from the isoprene, α-pinene SOA and β-caryophyllene SOA. Green circles indicating regions dominated by the monomers and dimers of α-pinene SOA are added to guide the eye.

**Minor comments:**

**1. Line 91. Change "outside" to "out".**

Accepted.

**2. Line 407 – 408. Some evidence to support this statement should be added.**

We write in lines 405 – 409 in the revised manuscript: "During the last 2–3 h of dark aging, the nitrate group fraction is stable, but photolysis causes a slight decrease of the nitrate-to-monomer ratio for all systems, consistent with the decrease of 1–3 % of the mass fraction of ONs of total organic compounds for all systems. It is clear from this that UV light fragmentation of nitrate groups only plays a minor role in changing the chemical composition of SOA when transitioning from dark to light conditions."

Based on this statement, lines 418 – 420 were revised as "However, based on the stability of the nitrate group fraction during photolytic aging as described above (compare also Fig. 5), the cleavage of the nitrate functional group from the carbon chain is not the main loss pathway in our study and on the time scales of our experiments."

**References**

Bell, D. M., Wu, C., Bertrand, A., Graham, E., Schoonbaert, J., Giannoukos, S., Baltensperger, U., Prevot, A. S. H., Riipinen, I., El Haddad, I., and Mohr, C.: Particle-phase processing of α-pinene NO$_3$ secondary organic aerosol in the dark, Atmos. Chem. Phys. Discuss., 2021, 1-28, 10.5194/acp-2021-379, 2021.

Buchholz, A., Ylisirniö, A., Huang, W., Mohr, C., Canagaratna, M., Worsnop, D. R., Schobesberger, S., and Virtanen, A.: Deconvolution of FIGAERO–CIMS thermal desorption profiles using positive matrix factorisation to identify chemical and physical processes during particle evaporation, Atmos. Chem. Phys., 20, 7693-7716, 10.5194/acp-20-7693-2020, 2020.

Donahue, N. M., Robinson, A. L., Stanier, C. O., and Pandis, S. N.: Coupled Partitioning, Dilution, and Chemical Aging of Semivolatile Organics, Environmental Science & Technology, 40, 2635-2643, 10.1021/es052297c, 2006.

Goracci, L., Tortorella, S., Tiberi, P., Pellegrino, R. M., Di Veroli, A., Valeri, A., and Cruciani, G.: Lipostar, a Comprehensive Platform-Neutral Cheminformatics Tool for Lipidomics, Analytical Chemistry, 89, 6257-6264, 10.1021/acs.analchem.7b01259, 2017.

Isaacman-VanWertz, G., and Aumont, B.: Impact of organic molecular structure on the estimation of atmospherically relevant physicochemical parameters, Atmos. Chem. Phys., 21, 6541-6563, 10.5194/acp-21-6541-2021, 2021.

James, P. F., Perugini, M. A., and O'Hair, R. A. J.: Sources of artefacts in the electrospray ionization mass spectra of saturated diacylglycerophosphocholines: From condensed phase hydrolysis reactions through to gas phase intercluster reactions, Journal of the American Society for Mass Spectrometry, 17, 384-394, 10.1021/jasms.8b02630, 2006.

Keith-Roach, M. J.: A review of recent trends in electrospray ionisation–mass spectrometry for the analysis of metal–organic ligand complexes, Analytica Chimica Acta, 678, 140-148, https://doi.org/10.1016/j.aca.2010.08.023, 2010.

Kourtchev, I., Szeto, P., O'Connor, I., Popoola, O. A. M., Maenhaut, W., Wenger, J., and Kalberer, M.: Comparison of Heated Electrospray Ionization and Nanoelectrospray Ionization Sources Coupled to Ultra-High-Resolution Mass Spectrometry for Analysis of Highly Complex Atmospheric Aerosol Samples, Analytical Chemistry, 92, 8396-8403, 10.1021/acs.analchem.0c00971, 2020.

Li, Y., Pöschl, U., and Shiraiwa, M.: Molecular corridors and parameterizations of volatility in the chemical evolution of organic aerosols, Atmos. Chem. Phys., 16, 3327-3344, 10.5194/acp-16-3327-2016, 2016.

Maire, F., and Lange, C. M.: Formation of unexpected ions from a first-generation polyamidoamine dendrimer by use of methanol: an artefact due to electrospray emitter corrosion?, Rapid Communications in Mass Spectrometry, 24, 995-1000, https://doi.org/10.1002/rcm.4475, 2010.

Mohr, C., Thornton, J. A., Heitto, A., Lopez-Hilfiker, F. D., Lutz, A., Riipinen, I., Hong, J., Donahue, N. M., Hallquist, M., Petäjä, T., Kulmala, M., and Yli-Juuti, T.: Molecular identification of organic vapors driving atmospheric nanoparticle growth, Nature Communications, 10, 4442, 10.1038/s41467-019-12473-2, 2019.

Nah, T., Sanchez, J., Boyd, C. M., and Ng, N. L.: Photochemical Aging of α-pinene and β-pinene Secondary Organic Aerosol formed from Nitrate Radical Oxidation, Environmental Science & Technology, 50, 222-231, 10.1021/acs.est.5b04594, 2016.

Peräkylä, O., Riva, M., Heikkinen, L., Quéléver, L., Roldin, P., and Ehn, M.: Experimental investigation into the volatilities of highly oxygenated organic molecules (HOMs), Atmos. Chem. Phys., 20, 649-669, 10.5194/acp-20-649-2020, 2020.

Rovelli, G., Jacobs, M. I., Willis, M. D., Rapf, R. J., Prophet, A. M., and Wilson, K. R.: A critical analysis of electrospray techniques for the determination of accelerated rates and mechanisms of chemical reactions in droplets, Chemical Science, 11, 13026-13043, 10.1039/D0SC04611F, 2020.

Stark, H., Yatavelli, R. L. N., Thompson, S. L., Kang, H., Krechmer, J. E., Kimmel, J. R., Palm, B. B., Hu, W., Hayes, P. L., Day, D. A., Campuzano-Jost, P., Canagaratna, M. R., Jayne, J. T., Worsnop, D. R., and Jimenez, J. L.: Impact of Thermal Decomposition on Thermal Desorption Instruments: Advantage of Thermogram Analysis for Quantifying Volatility Distributions of Organic Species, Environmental Science & Technology, 51, 8491-8500, 10.1021/acs.est.7b00160, 2017.

Takeuchi, M., and Ng, N. L.: Chemical composition and hydrolysis of organic nitrate aerosol formed from hydroxyl and nitrate radical oxidation of α-pinene and β-pinene, Atmos. Chem. Phys., 19, 12749-12766, 10.5194/acp-19-12749-2019, 2019.

Wu, R., Vereecken, L., Tsiligiannis, E., Kang, S., Albrecht, S. R., Hantschke, L., Zhao, D., Novelli, A., Fuchs, H., Tillmann, R., Hohaus, T., Carlsson, P. T. M., Shenolikar, J., Bernard, F., Crowley, J. N., Fry, J. L., Brownwood, B., Thornton, J. A., Brown, S. S., Kiendler-Scharr, A., Wahner, A., Hallquist, M., and Mentel, T. F.: Molecular composition and volatility of multi-generation products formed from isoprene oxidation by nitrate radical, Atmos. Chem. Phys., 21, 10799-10824, 10.5194/acp-21-10799-2021, 2021.

Zhao, D., Pullinen, I., Fuchs, H., Schrade, S., Wu, R., Acir, I. H., Tillmann, R., Rohrer, F., Wildt, J., Guo, Y., Kiendler-Scharr, A., Wahner, A., Kang, S., Vereecken, L., and Mentel, T. F.: Highly oxygenated organic molecule (HOM) formation in the isoprene oxidation by NO3 radical, Atmos. Chem. Phys., 21, 9681-9704, 10.5194/acp-21-9681-2021, 2021.

---

## Author Comment (AC2)

Response Letter to Referee #2

The authors thank the reviewer for the careful review of our manuscript and the helpful comments and suggestions. All the comments (in bold text) are addressed below point by point, with our response following in non-bold text and the corresponding revisions to the manuscript in blue. All updates to the original submission are tracked in the revised version.

**General Comments**

**This manuscript (which is a companion paper to one submitted by Bell et al.) describes results of a laboratory study of the effect of light on the mass and composition of SOA formed from the reaction of NO3 radicals with three terpenes: isoprene, a-pinene, and b-caryophyllene. Experiments were conducted in a Teflon chamber, SOA mass and size were monitored with an SMPS, and gas and particle composition were monitored with a FIGAERO-CIMS and EESI-TOF. Particle volatility was also estimated using a variety of SARs. Experimental evidence is presented indicating changes in the composition of the aerosol when the lights are on, with shifts from dimers to monomers, but with little corresponding evaporation. The observations are thoroughly discussed, and various possible explanations are proposed. In general, however, given the complexity of the system, the lack of information on the molecular structures of the SOA components (only elemental formulas are available), and the non-quantitative MS analyses, it was not possible to draw convincing conclusions about the mechanisms by which light might have altered the SOA. Nonetheless, the data set is interesting, and future studies may provide more detailed data that can help to explain the results. I think the manuscript can be published after the following comments are addressed.**

We thank the reviewer for the positive assessment of the manuscript.

**Specific Comments**

1. **Since neither the EESI-TOF or the FIGAERO-CIMS signals have been calibrated, the authors cannot assume that all compounds have the same sensitivity. All discussion about "mass changes" or "mass fractions" should therefore be changed to "signal changes" of signal fractions". These problems with the MS methods may help to explain why the changes in mass measured by the SMPS (a real mass measurement) are so much smaller than the changes inferred from the MS signals.**

Thank you for the comment. In the revised manuscript, all discussion about "mass changes" and "mass fractions" related to MS data is changed to "signal changes" and "signal fractions": Lines 24 – 25 in the revised manuscript were revised as "Overall, 48 %, 44 %, and 60 % of the total signal for the isoprene, α-pinene, and β-caryophyllene BSOA$_{NO3}$ was sensitive to photolytic aging and exhibited decay"; Lines 378 – 379 were revised as "These compounds contributed 48 %, 44 % and 60 % to the total pre-photolysis signal for these three systems."; Line 584 – 586 were revised as "Despite finding 0–12 % of the mass evaporating during photolysis, 48 %, 44 % and 60 % of the total signal of the isoprene, α-pinene, and β-caryophyllene SOA, respectively, is sensitive to photolytic aging, representing a majority of the pre-photolysis composition.".

2. **I find the plots in Figures 2 and 3 difficult to interpret. I would like to see similar plots for samples collected a few minutes apart to see how well the subtraction approach works. For such a comparison the spectra should essentially cancel out, giving a reader more confidence that what is shown in Figures 2 and 3 is not just statistical noise.**

With the FIGAERO-CIMS set up to measure continuously during the experiments, sample collection (10 – 20 min for most of the filters) was directly followed by thermal desorption, and the heating cycle took about 50 min. A comparison of samples only a few minutes apart is therefore not possible. We have added Fig. S7 to the supplementary information (SI) of our manuscript, where we plot the mass spectra of three filters (Filter 2, Filter 3 and Filter 4) collected during the 2 – 4.5 h of the dark aging period, and of Filter 5 collected during the photolysis period in Exp.3 ($\alpha$-pinene + $NO_3$) as an example. Filter 1 collected at the beginning of the experiment was excluded because most of the changes in chemical composition occur over the first 2 h of dark aging (Bell et al., 2021). The changes in chemical composition between the three filters during dark aging (Filter 2, Filter 3 and Filter 4) are much smaller compared to the changes between Filter 4 (Pre 2) and Filter 5 collected after photolysis (Post), thus the changes of the chemical composition during photolytic aging is not just statistical noise.

In the revised manuscript, in order to see the small changes during dark aging, we keep Fig. 3 as it is. We have added Fig. S7 to the SI and the following sentence to lines 360 – 361 of the revised manuscript: "Fig S7 in the SI shows the changes in absolute signal fraction for the filter samples pre and post photolysis for Exp. 3."

[Figure]

Fig. S7 Mass spectra of four filter samples collected during dark aging and photolysis in Exp.3.

3. **Similar to Comment 2, throughout the manuscript the authors discuss changes in signals on the order of 10% as if they are real. What evidence do they have for this? Have these experiments been replicated?**

In Fig. 3 and lines 362 – 370 of the revised manuscript, we report the changes in signal fraction of some major compounds in the three systems, with maximal changes in $C_{20}H_{32}N_2O_9$ (about 10 % decay in signal fraction). The changes in absolute signals of individual compounds were much larger. As mentioned in lines 374 – 378 of the revised manuscript, the average decay of the photolabile compounds in 1h was 44 % ± 20 %, 64 % ± 24 %, and 24 % ± 18 % for the

isoprene, α-pinene and β-caryophyllene SOA, respectively. Take $C_{10}H_{17}N_3O_{13}$ from the isoprene + $NO_3$ system as an example, its change in signal fraction was only 2 %, but the absolute signal significantly decayed by about 50 % (Fig. S8 in the revised SI).

Exp. 1 and Exp. 2 could serve as replicates. We observed similar chemical composition (Fig. 2a vs. Fig. S6) and also similar changes due to photolytic aging in these two experiments. 52 and 56 out of 359 compounds decreased significantly during photolysis, and they contributed 49.6 % and 47.2 % to the total pre-photolysis signal in Exp. 1 and Exp.2 respectively. The average decay of the photolabile compounds in 1h was 36 % ± 22 % and 52 % ± 19 % in Exp. 1 and Exp. 2 respectively.

4. **Line 455: I do not understand why partitioning cannot be a significant part of the explanation for the observed changes. If many of the oligomers are formed and dissociate by reversible reactions then this seems quite possible.**

We apologize we did not make this more clear in the text. By "repartitioning" here we mean evaporation of compounds from the particle phase to re-establish equilibrium that was disturbed due to photodegradation in the gas phase. Lines 465 – 469 were revised as "If gas-phase photo-degradation was the dominant cause for mass loss in all systems, then the largest decays in the particle phase would be expected from the most volatile species. Further, large non-volatile molecules (e.g. dimers in the β-caryophyllene SOA) should be non-responsive to such a pathway. Because there is a systematic degradation of dimers in all systems, it is unlikely that repartitioning derived from gas-phase photodegradation is driving the change in SOA composition during UV aging.".

It is worth mentioning that dissociation of particle-phase dimers by reversible reactions independent of light conditions (if this is what the reviewer meant) is not an important reason for the changes in chemical composition during photolysis in our study. In Fig.3, we compared firstly the two filters during dark aging to check for reactions, including reversible dimerization, that are not necessarily associated with photolysis. The changes during dark aging were much smaller than the changes due to photolysis, and more importantly different compared to the changes due to photolysis. The signal fractions of smaller compounds decreased and that those of bigger compounds increased during dark aging, which is opposite to what happened during photolysis. Thus, we can exclude reversible dimerization independent of light conditions as a possible explanation.

5. **Line 480: Since the major sink for OH formed by photolysis in these experiments is reaction with NO2, then a significant amount of HNO3 is formed. Couldn't this HNO3 catalyze the decomposition of dimers, helping to explain observed MS changes?**

In our experiments, the gas-phase $HNO_3$ was very high even during dark aging, likely stemming from hydrolysis of $N_2O_5$ in our injection inlet or on chamber walls. Switching on the lights did not cause significant changes (< 10 % of pre-photolysis signal in all experiments) in the gas-phase $HNO_3$ signal (see an example Exp. 2 in Fig. R1). As mentioned above, during dark aging, the signal fractions of smaller compounds decreased and that those of bigger

compounds increased, thus decomposition of dimers catalysed by $HNO_3$ was unlikely important for the changes in chemical composition during dark aging. Therefore, it also cannot explain the changes we observed during photolysis. In the revised manuscript, we have added the following sentence to line 495: "However, we didn't observe significant increase in the gas-phase $HNO_3$ signal."

[Figure]

Fig. R1 Normalized signal of gas-phase $HNO_3I^-$ before and after photolysis from Exp. 2 (isoprene + $NO_3$).

**Technical Comments**

**None.**

**References**

Bell, D. M., Wu, C., Bertrand, A., Graham, E., Schoonbaert, J., Giannoukos, S., Baltensperger, U., Prevot, A. S. H., Riipinen, I., El Haddad, I., and Mohr, C.: Particle-phase processing of α-pinene $NO_3$ secondary organic aerosol in the dark, Atmos. Chem. Phys. Discuss., 2021, 1-28, 10.5194/acp-2021-379, 2021.

---

## Author Response (AR1)

**Response Letter to Referee #1**

The authors thank the reviewer for the careful review of our manuscript and helpful comments and suggestions. All the comments (in bold text) are addressed below point by point, with our response following in non-bold text and the corresponding revisions to the manuscript in blue. All updates of the original submission are tracked in the revised version.

**Overall comment:**

This work examined the composition and volatility changes of SOA formed from NO3 oxidation of three biogenic VOCs upon UV photolysis. This study used two state-of-theart SOA molecular analysis instruments, FIGAERO-CIMS and EESI-TOF to examine the SOA composition change before and after photolysis. The authors also compared a few methods that were used to estimate SOA bulk volatility. The SOA compositions were shown to be very interesting that the oligomer fractions were extremely high (higher than ever reported for SOA systems) for all the three biogenic VOCs. The comparisons of bulk volatility estimates lead to a conclusion that SOA that largely contain nitrate groups are highly uncertain. Overall, the manuscript was well written, and the experiments and analyses were carefully carried out. However, there are some major concerns regarding the interpretation of the instruments and chemistry in the NO3 oxidation. These concerns should be addressed before the manuscript can be published in ACP.

We thank the reviewer for the positive assessment of our manuscript.

**Major comments:**

1. Line 134 – 139. The determination of negligible fragmentation fractions due to thermal desorption is unclear. For a certain m/z or fitted chemical formula, FIGAERO-CIMS can usually produce an overall thermogram with a broad peak or peaks, each of which can be integrated by a number of individual species' thermogram peaks (Schobesberger et al., 2018, ACP). Deconvoluting the individual peaks might involve large uncertainties. When there is more than one broad peaks, it is apparent that the later broad peak is from thermal decomposition of a much lower volatility compound such as dimers (e.g., Figure S2a). But thermal decomposition peaks can also be hidden under the same broad peak, especially when the parent compound and the fragmentation compound have more close volatilities. For example, assuming a compound with formula CxHyOzN undergo thermal decomposition (dehydration) and produces CxHy-2Oz-1N, the decomposition product could be difficult to separate from original CxHy-2Oz-1N in the broad thermogram peak. The two steps described in the supporting information may not be able to identify such thermal decomposition. Therefore, the percentage ranges provided in Line 134–135 are likely the lower estimate of the thermally labile species in these SOA.

We thank the reviewer for pointing out this important issue. However, what needs to be compared is the *thermal decomposition temperature* of compound  $C_xH_yO_zN$  forming e.g.  $C_xH_{y-2}O_{z-1}N$  (dehydration) or  $C_{x-1}H_yO_{z-2}N$  (decarboxylation) with the *thermal desorption temperature* of compounds with the same formula as the thermal fragmentation products. Dehydration or decarboxylation reactions are expected to occur at temperatures higher than 120 °C (Buchholz et al., 2020; Stark et al., 2017), while the thermal desorption temperatures of major compounds in the isoprene and  $\alpha$ -pinene systems and the monomers in the  $\beta$ caryophyllene system are below 120 °C. As shown in Fig. S3 (which is added as a new figure to the supplementary information (SI) of our manuscript) below, most of the top 30 compounds in the isoprene and  $\alpha$ -pinene systems and major monomers in the  $\beta$ -caryophyllene system do not have "shoulders" or "plateaus", and their signals in the temperature range higher than 120 °C are relatively low. Therefore, for these compounds, signals from desorbing compounds and thermal fragmentation products are well separated. There are a few exceptions with bimodal thermograms (Fig. S3c, blue lines), which were corrected as described in the manuscript in the SI S1.2. Even though for some compounds with thermal desorption temperature higher than 120 °C (e.g. the dimers in the  $\beta$ -caryophyllene system, Fig. S3c&d), interference from fragmentation may be possible as mentioned by the reviewer, it is rather unlikely an issue in our study. The molecular formulae of the compounds with significant tailings (Fig. S3d) do not correspond to thermal decomposition products of major dimers (e.g. dehydration products with m/z - 18.015 compared to the parent compounds), but the tailings might rather be signal from isomers of these major dimers. Moreover, the major peaks observed by the FIGAERO-CIMS and the EESI-TOF, for which no thermal fragmentation can occur, are similar. For example, for the isoprene system, the dominating compound measured by both instruments is  $C_{10}H_{17}N_3O_{13}$ , and no substantial signal of  $C_{10}H_{15}N_3O_{12}$  (m/z –18.015) is found.

We have added Fig. S3 to the SI of our manuscript, together with explanations based on the above response (S1.2): "Fig. S3 shows thermal fragmentation to be a minor issue in our study. Most of the peaks in the three systems do not have "shoulders" or "plateaus". In addition, for almost all compounds in the isoprene and  $\alpha$ -pinene systems and the monomers in the  $\beta$ caryophyllene system, the signals in the temperature range higher than 120 °C where dehydration or decarboxylation reactions are expected to occur (Buchholz et al., 2020; Stark et al., 2017) are relatively low. There are a few exceptions with bi-modal thermograms (Fig. S3c, blue lines), which were corrected for as described above. Regarding the dimers from the βcaryophyllene system (Figure S3d), there are a few compounds with significant tailings. However, their molecular formulae do not correspond to thermal decomposition products of major dimers (e.g. dehydration products with m/z - 18.015 compared to the parent compounds), but the tailings might rather be signal from isomers of these major dimers." Lines 135 – 141 in the revised manuscript were rewritten as following: "As shown in Fig. S3, most of the thermograms of individual compounds for all systems exhibited unimodal and sharp peaks. Thermal fragmentation products were detected either through thermograms of individual compounds with multiple peaks (normally double peaks) or one peak with  $T_{max}$  (desorption temperature at which a compound's signal exhibits a maximum) much higher than the estimated  $T_{max}$  (e.g., for  $\alpha$ -pinene SOA, a C10 monomer had a  $T_{max} \approx 140$  °C which is in the range of  $T_{max}$  for C20 dimers (Faxon et al., 2018)). Artefacts resulting from thermal fragmentation products were corrected for (see the SI S1.2 and Fig. S2)."

Fig. S3 Thermograms of top 30 (signal) compounds of the SOA from (a) Exp.1, (b) Exp. 3 and (c) Exp. 4 and (d) major dimers from Exp. 4.

2. This is a related questions regarding thermal decomposition. In  $NO_3$  + biogenic VOC oxidation systems, it is likely common to have PAN species from  $RO_2$  +  $NO_2$  (NO2 as a byproduct from N2O5 decomposition). PAN species are very thermally labile and produce  $RO_2$  +  $NO_2$ . This may happen even under moderate heating temperatures. Do the authors have a clue to what extent this chemistry has happened in FIGAERO-CIMS? This class of species was not mentioned throughout the manuscript.

We thank the reviewer for bringing up an interesting point. Since the information that can be obtained from mass spectrometers is molecular composition, but not molecular structure, we are limited in assigning the detected compounds to a certain group of compounds such as PANs. As the reviewer pointed out, PAN species could be formed in our system through RO2 + NO2, however, as oligomers are dominating in our isoprene and  $\alpha$ -pinene systems, this termination pathway which produces monomer compounds is unlikely as important as the RO2 + RO2 pathway. For example in the  $\alpha$ -pinene system, we observed monomers with molecular formulae corresponding to pinonaldehyde-PAN (C10H15NO6) and norpinonaldehyde-PAN (C9H13NO6) (Nah et al., 2016), but their signals were much lower than that of the dominating dimer C20H32N2O9 (Fig. 2). For the  $\beta$ -caryophyllene system, we did observe a big fraction of monomers and some of them have PAN-like molecular formulae, e.g. C15H23NO7,8. However, overall, the sum of signals of C15HyOzN is 25 times higher than the sum of signals of C15HyOz.

for all three SOA systems, we would observe the thermal decomposition products as CHO (without nitrate groups). Since CHO compounds only contribute a very small fraction to the total signal, thermal decomposition of PAN species unlikely plays a significant role for the SOA generated in our study. In addition assuming thermal decomposition would play a lesser role in the EESI-TOF compared to the FIGAERO-CIMS, more potential fragmentation products and fewer compounds with molecular formulae corresponding to PANs would be expected in the FIGAERO-CIMS mass spectra. By comparing the molecular compositions detected by the FIGAERO-CIMS and the EESI-TOF, we did not observe big differences in major compounds measured by two instruments.

**3.** Line 177. The authors clarified how particle mass concentration was corrected for wall loss and coagulation. But it is unclear how the ion signals from FIGAERO-CIMS and EESI-TOF were corrected, especially for coagulation. Can the detail be elaborated?**

We apologize, this was not correctly stated in the manuscript. As the signal of the FIGAERO-CIMS and the EESI-TOF is based on particle mass and not particle number, no correction for coagulation is needed. Lines 178 - 184 were revised as following to clarify the correction of the FIGAERO-CIMS and EESI-TOF signal for wall loss (but not for coagulation): "They were corrected for wall losses using a uniform dynamic wall loss rate  $k_{wall}$  for the whole size range.  $k_{wall}$  was determined from the observed exponential decay of the particle number concentration (taking coagulation into account) using Eq. 2, where N corresponds to the particle number concentration and  $k_{coag}$  corresponds to the coagulation coefficient (5\*10-10 s-1 (Pospisilova et al., 2020)).

$$\frac{dN}{dt} = -k_{\text{coag}} * N^2 - k_{\text{wall}} * N \tag{Eq. 2}$$

The wall loss-corrected particle mass concentration was divided by the uncorrected mass concentration to obtain the wall loss correction factor applied to the EESI-TOF and FIGAERO-CIMS signal."

4. The reported oligomer fractions are very high in this work. In fact, I have not seen any studies reporting such high fractions of oligomers and wonder what could have caused these results. For the a-pinene + NO3 system, Nah et al. (2016 ES&T) found that the majority of the SOA was made by monomers. The discrepancy could be due to two different conditions: (i) Nah et al. used much lower VOC (~ 10 ppb) vs. 100 ppb in this work; and (ii) Nah et al. had RO2 + HO2 as the dominant condition vs. RO2+RO2 in this work; But even these differences are responsible for the large oligomer fractions reported in this work, how is it possible that monomer products from RO2 + RO2 did not partition and form SOA? The SOA mass loadings were still > 10 ug/m3. Some discussion or explanation should be provided.

For the isoprene+NO3 system, it is not surprising that the oligomers are dominating, because the monomers are too volatile. For a typical C5 compound with 2 nitrate groups, e.g.  $C_5H_8N_2O_8$ , which is the dominating gas-phase monomer observed in our study (and has been reported in other studies, e.g. Zhao et al. (2021)), the estimated  $log_{10}C^*$  at 300K is between 1.53 and 4.4 based on the three parameterizations used in this study (Mohr et al., 2019; Peräkylä et al., 2020; Isaacman-VanWertz and Aumont, 2021), i.e., in the SVOC range (Donahue et al., 2006).

For the  $\alpha$ -pinene + NO3 and  $\beta$ -caryophyllene + NO3 systems, we did observe monomers (see Fig. 2 and also new SI figure Fig. S7), and the large contributions from oligomers can be explained by the dominating RO2 + RO2 and RO2 + NO3 pathways. In a more similar study to ours of  $\alpha$ -pinene + NO3 SOA, in which reactions between peroxy radicals (RO2 + RO2) and RO2 + NO3 are favoured (Takeuchi and Ng, 2019), the SOA measured by the FIGAERO-CIMS was also dominated by C20 dimers. Their dominating dimer was C20H32N2O9 and the dominating monomer was C10H15NO6, identical to our study. Their ratio of C20H32N2O9 to C10H15NO6 was about 5, in our study ~ 2 times higher (about 10) (Fig. 2). Therefore, the large fraction of oligomers shown here are not so surprising, and likely come from the large concentrations of RO2 radicals present.

In the revised manuscript, we have added this discussion after mentioning the study of Takeuchi and Ng (2019). Lines 302 - 307 were revised as following: "Our mass spectra are similar to those reported in Takeuchi and Ng (2019) measured with a FIGAERO-CIMS, where the  $\alpha$ -pinene + NO3 SOA was dominated by C20 dimers with two dominating compounds, C20H32N2O9 and C20H32N2O10. The dominating monomer in their study was C10H15NO6, also identical to our study. Their ratio of C20H32N2O9 to C10H15NO6 was about 5, in our study ~ 2 times higher (about 10). In the system in Takeuchi and Ng (2019), reactions between peroxy radicals (RO2 + RO2) and RO2 + NO3 are also favoured. Large concentrations of RO2 present in both studies likely lead to the large fraction of oligomers."

**5. Line 304 and 310. Can you elaborate why CHO compounds were more prominent in EESI-TOF and what "degradation pathways" were referred to? In principle, the FIGAERO-CIMS requires heating and should be more likely to decompose CHON compounds than the EESI-TOF.**

We are currently preparing a manuscript about the potential artefacts of the EESI-TOF. Briefly, in our NO3-initiated systems with a high abundancy of dinitrate dimer species (or oligomers with an even higher number of nitrate groups),  $C_xH_yO_z$  species that were observed by the EESI-TOF but not by the FIGAERO-CIMS such as e.g.  $C_4H_6O_2$  (mentioned in line 292),  $C_{10}H_{16}O_3$  and  $C_{10}H_{16}O_4$  (mentioned in line 311) are likely related to artefacts from the electrospray ionization in the EESI-TOF (Goracci et al., 2017; James et al., 2006; Keith-Roach, 2010; Maire and Lange, 2010; Kourtchev et al., 2020; Rovelli et al., 2020). Details are mentioned in lines 328 - 330.

For clarification reasons we have rearranged the discussion on artefacts in the EESI-TOF in the revised manuscript as following: In lines 293 - 294, we deleted "to be discussed in a future paper (Bell et al. in preparation)". In lines 311 - 312, we deleted "which may also stem from some degradation pathways in the EESI-TOF source". Lines 324 - 330 were revised as following: "Differences are also caused by the fragmentation of oligomers into smaller compounds (CxHyOz) inside the EESI-TOF, such as C4H6O2 in the isoprene + NO3 system, and a few CHO compounds in the  $\alpha$ -pinene + NO3 and  $\beta$ -caryophyllene + NO3 systems. These compounds are likely related to artefacts from the electrospray ionization (Goracci et al., 2017; James et al., 2006; Keith-Roach, 2010; Maire and Lange, 2010; Kourtchev et al., 2020; Rovelli

et al., 2020). They may occur due to the proximity of  $-ONO_2$  functional groups next to peroxy linkages which results in fragmentation when exposed to water in the electrospray (Bell et al, in preparation), and a loss of a HNO3 fragment during ionization (Liu et al., 2019)."

**6. Can the authors show the changes of particle-phase $N_2O_5$ and $HNO_3$ signals (FIGAERO-CIMS) under the various conditions? Since $N_2O_5$ is likely in excess and the experiments were carried out under humid conditions, it is interesting to see how $N_2O_5$ uptake might have affected the chemistry. In fact, the chemical compositions shown in Figure 2 have many peaks with 3-4 N in the formulas. Could they have to do with particle-phase chemistry related to $N_2O_5$ uptake, rather than just gas-phase oxidation?**

For more details on the role of  $N_2O_5$  during the dark aging period, we refer the reader to the companion paper by Bell et al. (2021),which is available online (https://acp.copernicus.org/preprints/acp-2021-379/). Briefly, with the FIGAERO-CIMS we observed a small fraction of  $N_2O_5$  present in the particle phase, which could act as an oxidant. The fraction of  $N_2O_5$  in the particle phase decreased with time during the experiment (by ~50%) over  $\sim 3$  h, shown in Figure S8 in Bell et al. (2021)), indicating it was either consumed or evaporating from the particle phase. Overall,  $N_2O_5$  or peroxide-containing organics present in the particle phase could be responsible for the observed continued oxidation, most of which occurred over the first 2 h of dark aging, however.

For the particle-phase HNO3, as the gas-phase HNO3 signal was very high, the particle-phase signal was impacted by the gas-phase measurement. Thus, it is not possible to quantify the changes between the different samples.

7. SOA volatility. The authors compared a number of methods to estimate SOA volatilities and the results are very different. The authors concluded that the molecular formula derived parameterization is highly uncertain, especially for the compounds with multiple nitrate groups. Can the authors break down the comparison by #N? For example, how different are the methods for species with 0-4 nitrogen atoms? Also, dimers with multiple N atoms were found to have similar Tmax as monomers. Could this be real or some unrecognized artifact in FIGAERO-CIMS? If this were true, it somewhat contradicts the earlier discussions regarding evaporation of SOA after photolysis.

We thank the reviewer for these suggestions. In fact we have made a new figure where we show  $\log_{10}C^*$  based on the parameterizations of Mohr et al. (2019), Li et al. (2016) (updated by Isaacman-VanWertz and Aumont (2021)), and Peräkylä et al. (2020) of major compounds with 1 - 4 N from all three systems (Fig. S10 in the SI of the revised manuscript). In the small panel of Fig. S10, only C10 compounds are compared in order to exclude the influence by different carbon numbers. It is apparent that the difference between the  $\log_{10}C^*$  from the three parameterizations becomes larger with increasing nitrogen number. Similar comparison study performed by Wu et al. (2021) showed as well that the discrepancy in saturation vapor pressure from different parametrizations increases with increasing complexity of molecules. We have added the following sentence to lines 513 - 514 of the revised manuscript: "As Fig. S10 in the SI shows, the discrepancy of  $\log_{10}C^*$  from the three parameterizations becomes larger with increasing nitrogen number:"